# Minimax Optimal Quantile and Semi-Adversarial Regret via Root-Logarithmic Regularizers

**Jeffrey Negrea**[*]
University of Toronto
jeffrey.negrea@mail.utoronto.ca

**Blair Bilodeau**[*]
University of Toronto
blair.bilodeau@mail.utoronto.ca

**Nicolò Campolongo**
Spanflug Technologies GmbH
nico.campolongo@spanflug.de

**Francesco Orabona**
Boston University
francesco@orabona.com

**Daniel M. Roy**
University of Toronto
daniel.roy@utoronto.ca

## Abstract

Quantile (and, more generally, KL) regret bounds, such as those achieved by NormalHedge (Chaudhuri, Freund, and Hsu 2009) and its variants, relax the goal of competing against the best individual expert to only competing against a majority of experts on adversarial data. More recently, the semi-adversarial paradigm (Bilodeau, Negrea, and Roy 2020) provides an alternative relaxation of adversarial online learning by considering data that may be neither fully adversarial nor stochastic (i.i.d.). We achieve the minimax optimal regret in both paradigms using FTRL with separate, novel, root-logarithmic regularizers, both of which can be interpreted as yielding variants of NormalHedge. We extend existing KL regret upper bounds, which hold uniformly over target distributions, to possibly uncountable expert classes with arbitrary priors; provide the first full-information lower bounds for quantile regret on finite expert classes (which are tight); and provide an adaptively minimax optimal algorithm for the semi-adversarial paradigm that adapts to the true, unknown constraint faster, leading to uniformly improved regret bounds over existing methods.

## 1  Introduction

We focus on the setting of learning with expert advice [V90; LW94], where in each round the learner selects a probability distribution over experts, observes the loss of each expert, and incurs the average loss of the experts under the learner's selected distribution. The learner's objective is to minimize regret against some mixture of the experts, which is the difference between their cumulative loss and the cumulative loss of the expert mixture over $T$ rounds.

The classical "worst-case" online learning paradigm assumes that the losses are adversarial—that is, they are chosen to make the learner perform as poorly as possible—and demands that the learner competes against the best-performing expert. However, there are many real-world settings where this assumption is too pessimistic, and consequently we focus on designing algorithms with provable guarantees that adapt to easier notions of both data and performance measures. A non-exhaustive list

---

[*]Equal contribution authors.

35th Conference on Neural Information Processing Systems (NeurIPS 2021).

of work on "easy data" includes Freund and Schapire [FS97], Cesa-Bianchi, Mansour, and Stoltz [CMS07], van Erven, Grünwald, Koolen, and de Rooij [v+11], and Gaillard, Stoltz, and van Erven [GSv14], all of which use variants of the Hedge algorithm [FS97] to obtain regret bounds in terms of data-dependent quantities. Ideally, such quantities are small when the data are "easy" to predict.

In this work, we focus on two paradigms beyond the classical worst-case: first, we consider relaxing the performance measure to *quantile (KL) regret*, which measures the ability of an algorithm to compete against an unknown mixture of the experts that potentially performs worse than a point-mass on the single best expert, and second, we consider regret within the *semi-adversarial paradigm*, which defines a spectrum of constraints on the permissible data distributions between stochastic and adversarial. The concept of $\epsilon$-quantile regret was introduced by Chaudhuri, Freund, and Hsu [CFH09], in which the player competes against the $\lfloor \epsilon N \rfloor$ best experts (out of $N$ total) rather than the single best. The authors demonstrated empirically that Hedge does poorly in this paradigm, and introduced a new algorithm NormalHedge with an upper bound on quantile regret of $\sqrt{(T + (\log N)^2)(1 + \log(1/\epsilon))}$. Later algorithms improved it to $\sqrt{T(1 + \log(1/\epsilon))}$ [CV10; OP16], removing the dependency on $N$. The semi-adversarial paradigm considers constraining the adversary's choice of data distributions, which was first motivated by Rakhlin, Sridharan, and Tewari [RST11]. Bilodeau, Negrea, and Roy [BNR20] extended this idea, defining *adaptive* minimax regret with respect to such constraints and providing an efficient algorithm with corresponding regret bounds.

**Contributions** While the best known algorithms for the above two paradigms are intrinsically different, we show that the follow-the-regularized-leader (FTRL) algorithm with new root-logarithmic regularizers achieves minimax optimal performance for both quantile and semi-adversarial regret. First, we provide the first FTRL algorithm with minimax optimal quantile regret guarantees, and do so without using the additive normalization step of previous algorithms [CL06, Section 2.1]. We achieve root-KL bounds that hold uniformly over target distributions on (possibly uncountable) expert classes with arbitrary priors, and reduce to the optimal quantile regret for discrete uniform priors. Moreover, we prove matching lower bounds for quantile regret with finite expert classes, demonstrating the optimality of known upper bounds (including our own) for the first time. Finally, in the semi-adversarial paradigm, we improve the dependence on the number of experts in the regret bound, obtaining uniformly improved upper bounds over previous work.

We achieve the above results through a novel local-norm analysis of FTRL with *linearly decomposable* regularizers on general (possibly uncountable) expert spaces. We use this analysis in conjunction with basic conditions on the first and second derivatives to design and analyze the root-logarithmic regularizers. We believe that this approach is fundamentally different from existing ones and could lead to further advances in obtaining optimal algorithms. In fact, there exist results stating if a regret bound is achievable by some algorithm, then that same bound is nearly achievable by mirror descent with some potential function (see, e.g., Srebro, Sridharan, and Tewari [SST11, Thm 9]). However, it is not clear how to design such a function in practice. In contrast, our general FTRL bound reduces the choice of regularizer to a single univariate function, and clarifies how fundamental properties of this function lead to trade-offs in the regret bound.

**Related Work** Chernov and Vovk [CV10] first discussed the fact that the $\epsilon$-quantile regret corresponds to the KL divergence between a uniform prior and the uniform mixture of the top $\lfloor \epsilon N \rfloor$ competitors, and such KL bounds have consequently quickly followed quantile regret bounds. Luo and Schapire [LS14; LS15] provided a variant of NormalHedge along with a data-dependent KL regret bound at the cost of an additional $\log T$ factor; a similar but even tighter result was obtained by Koolen and van Erven [Kv15]. Orabona and Pál [OP16] showed that these algorithms can be obtained with a reduction to optimal coin-betting online algorithms, yet none of them can be reduced to an FTRL algorithm. Independently, Harvey, Liaw, Perkins, and Randhawa [H+20] proved that a similar strategy allows one to achieve the optimal anytime regret for the setting with $N = 2$ experts.

The first alternative to additively normalized algorithms in the (bandit) learning with experts setting was the INF algorithm [AB09], which was later recast as online mirror descent [ABL11]. Alquier [A21] also did not use additive normalization, and first introduced FTRL with $f$-divergences (focusing on the $\chi^2$-divergence) along with regret bounds for continuous distributions with certain unbounded loss functions. For the specific case of Hedge (and its online mirror descent analogue), an analysis on continuous spaces was given by Krichene, Balandat, Tomlin, and Bayen [K+15] and then more generally by van der Hoeven, van Erven, and Kotłowski [vvK18], while a coarser analysis for $f$-divergences was given by Alquier [A21]. Finally, choosing regularizers that are tuned to minimize

regret for specific tasks has recently led to advances in the online learning literature, both with bandit feedback [A+17; WL18; F+16] and in the full-information setting [LWZ18].

The best known asymptotic lower bound for learning with expert advice is by Cesa-Bianchi, Freund, Haussler, Helmbold, Schapire, and Warmuth [C+97], while a finite-time lower bound that asymptotically matches the leading constant was proved by Orabona and Pál [OP15a]. We are not aware of lower bounds for the quantile regret, with the notable exception of Koolen [K13], who proved a lower bound on the regret of learning with two experts based on the KL divergence against some prior.

## 2   Notation

To analyze FTRL beyond the finite setting requires some more care, and we rely on the language of measure theory to handle finite and uncountable expert classes simultaneously. Let $(\Theta, \Sigma)$ be a measurable space, $\mathcal{M}^{\infty}(\Theta, \Sigma)$ and $\mathcal{M}(\Theta, \Sigma)$ denote the collection of finite and probability measures respectively, and $\nu \in \mathcal{M}^{\infty}(\Theta, \Sigma)$ be arbitrary. A measure $q \in \mathcal{M}^{\infty}(\Theta, \Sigma)$ is *absolutely continuous* with respect to $\nu$ (denoted $q \ll \nu$) if $q(A) = 0$ for all $A \in \Sigma$ such that $\nu(A) = 0$. Let $\mathcal{M}_{\nu}(\Theta, \Sigma) = \{q \in \mathcal{M}(\Theta, \Sigma) : q \ll \nu\}$ be the set of probability measures that are absolutely continuous with respect to $\nu$. For any integer $N$, let $[N] = \{1, \ldots, N\}$, and $\mathrm{simp}([N]) = \{p \in [0,1]^N : \sum_{i=1}^{N} p_i = 1\}$. Let $\mathbb{R}_+ = [0, \infty)$ and $\mathbb{R}_{++} = (0, \infty)$, and define $\mathbb{M} \equiv \mathbb{M}((\Theta, \Sigma), (\mathbb{R}, \mathcal{R}))$ to be the space of measurable functions from $(\Theta, \Sigma)$ to $(\mathbb{R}, \mathcal{R})$, where $\mathcal{R}$ is the Borel $\sigma$-field on $\mathbb{R}$. Define sets of bounded measurable functions

$$\mathcal{L}^{\infty} = \{\ell \in \mathbb{M} : \sup_{\theta \in \Theta} |\ell(\theta)| < \infty\} \text{ and } \mathcal{L}_{[0,1]}^{\infty} = \{\ell \in \mathbb{M} : 0 \le \ell \le 1\},$$

sets of integrable functions

$$\mathcal{L}^1(\nu) = \{x \in \mathbb{M} : \int |x(\theta)| \, \nu(\mathrm{d}\theta) < \infty\} \text{ and } \mathcal{L}_+^1(\nu) = \{x \in \mathcal{L}^1(\nu) : x \ge 0\},$$

and the set of Radon–Nikodym derivatives (w.r.t. $\nu$) of probability measures

$$\mathcal{X}(\nu) = \{x \in \mathcal{L}_+^1(\nu) : \int x(\theta)\nu(\mathrm{d}\theta) = 1\}.$$

For every $x \in \mathcal{X}(\nu)$, let $\nu^{(x)} \in \mathcal{M}_{\nu}(\Theta, \Sigma)$ denote the unique probability measure satisfying $\mathrm{d}\nu^{(x)}/\mathrm{d}\nu = x$. For every $x, z \in \mathcal{L}^{\infty}$, let $\mathrm{Conv}(x, z) = \{\alpha x + (1 - \alpha)z : \alpha \in [0, 1]\}$ denote their convex hull. For every $\ell \in \mathcal{L}^{\infty}$ and $x \in \mathcal{L}^1(\nu)$, define

$$\langle \ell, \ x \rangle_{\nu} = \int \ell(\theta)x(\theta)\nu(\mathrm{d}\theta).$$

**Prediction with expert advice** We consider the following setting of online linear optimization. For each round $t \in [T]$, the player selects $\mu_t \in \mathcal{M}(\Theta, \Sigma)$ based only upon information available prior to round $t$, and then observes $\ell_t \in \mathcal{L}^{\infty}$. Performance is measured using the *regret* against some probability measure $q \in \mathcal{M}(\Theta, \Sigma)$, which is defined by

$$R_T(q) = \sum_{t=1}^{T} \mathbb{E}_{\theta \sim \mu_t} [\ell_t(\theta)] - \sum_{t=1}^{T} \mathbb{E}_{\theta \sim q} [\ell_t(\theta)].$$

The player's goal is to select $\mu_t$ so that the cumulative loss is not much larger than that of the average loss of an expert under $q$, and consequently to have small regret.

The elements of $\Theta$ can be regarded as experts for some prediction problem, and the learner aggregates the predictions of the experts by selecting an expert at random according to $\mu_t$ at round $t$. For concreteness, note that the usual prediction with expert advice setting for $N$ experts corresponds to $\mathcal{M}(\Theta, \Sigma) = \mathrm{simp}([N])$. Note that, for any convex loss, applying online linear optimization to the gradients of the losses provides an upper bound on the performance of online convex optimization.

**Follow-the-regularized-leader** Let $\overline{\mathbb{R}} = \mathbb{R} \cup \{+\infty\}$. Follow-the-regularized-leader (FTRL) [S07; AHR08; HK10] forms a broad class of algorithms for online convex optimization. For a finite measure $\nu \in \mathcal{M}^{\infty}(\Theta, \Sigma)$ and a sequence of regularizers $(\psi_t : \mathcal{X}(\nu) \to \overline{\mathbb{R}})_{t \in [T]}$, $(\psi_t)_{t \in [T]}$-regularized FTRL is defined by selecting $\mu_{t+1} = \nu^{(x_{t+1})}$ using

$$x_{t+1} \in \operatorname*{arg\,min}_{x \in \mathcal{X}(\nu)} \{\langle L_t, \ x \rangle_{\nu} + \psi_{t+1}(x)\}, \tag{1}$$

where $L_0 = 0$ and $L_t = \sum_{s=1}^{t} \ell_s$.

As mentioned above, a classical algorithm for prediction with expert advice on a finite class of $N$ experts is Hedge. While originally analyzed using potential functions, Hedge also corresponds to FTRL where $\nu$ is the counting measure and $\psi_t(\cdot) = -\eta_t^{-1} H(\cdot)$ for the Shannon entropy $H(x) = \sum_{i \in [N]} x(i) \log(1/x(i))$ with any sequence of regularizer scalings $(\eta_t)_{t \in \mathbb{N}} \subseteq \mathbb{R}_{++}$.

Depending on whether the space is continuous or discrete, the range of elements of $\mathcal{X}(\nu)$ will change, and consequently also the minimal domain on which regularizers must be defined. Let $\underline{\nu} = \inf \{\nu(A) : A \in \Sigma, \nu(A) > 0\}$, and let $\mathcal{S}(\nu) = [0, 1/\underline{\nu}]$ if $\underline{\nu} > 0$ and $\mathbb{R}_+$ otherwise. By definition, $\nu(\{\theta \in \Theta : x(\theta) \in \mathcal{S}(\nu)\}) = 1$ for all $x \in \mathcal{X}(\nu)$.

Concretely, when $\nu$ is counting measure then $\mathcal{S}(\nu) = [0, 1]$, when $\nu$ is uniform on $[N]$ then $\mathcal{S}(\nu) = [0, N]$, and when $\nu$ is a continuous distribution then $\mathcal{S}(\nu) = \mathbb{R}_+$. In this work, we consider *linearly decomposable* regularizers of the form $\psi_t(x) = \eta_t^{-1} \Psi_f^\nu(x)$, where

$$\Psi_f^\nu(x) = \int f(x(\theta)) \nu(\mathrm{d}\theta) \tag{2}$$

for some $f : \mathcal{S}(\nu) \to \mathbb{R}$. We refer to the algorithm that selects $\mu_{t+1}$ using Eq. (1) with a regularizer of the form in Eq. (2) as $\Psi_f^\nu$-regularized FTRL with regularizer scaling $\eta_t$.

## 3 Applications of FTRL with linearly decomposable regularizers

In Section 5, we provide a general analysis of FTRL with linearly decomposable regularizers for arbitrary $f$. First, we motivate such a general analysis by demonstrating the benefits of studying choices of $f$ beyond traditional FTRL regularizers with multiple examples, including quantile regret. To do so, we state the following corollaries of our general FTRL regret bound (Theorem 2 in Section 5) that achieve "root-KL" and variance bounds respectively. Proofs are deferred to Appendix B.

**Corollary 1.** *Suppose $\nu \in \mathcal{M}(\Theta, \Sigma)$ and $f : \mathbb{R}_+ \to \mathbb{R}$ satisfies:*

1. *$f(1) = 0$;*
2. *$f$ is twice continuously differentiable, $f'' > 0$ on $\mathbb{R}_{++}$, and $f''(0+) > 0$;*
3. *either $f$ or $1/f''$ is increasing on $\mathbb{R}_{++}$;*
4. *there exist $c_1, c_2 > 0$ such that $f'' \cdot (f + c_1) \geq c_2$.*

*For any sequence $(\ell_t)_{t \in \mathbb{N}} \subseteq \mathcal{L}_{[0,1]}^\infty$, $q \in \mathcal{M}_\nu(\Theta, \Sigma)$, and $T$, $\Psi_f^\nu$-regularized FTRL with regularizer scaling $\eta_t = \sqrt{c_2/t}$ achieves*

$$R_T(q) \leq \sqrt{\tfrac{T+1}{c_2}} \, \Psi_f^\nu(\mathrm{d}q/\mathrm{d}\nu) + \tfrac{c_1}{\sqrt{c_2}} \sqrt{T}.$$

The regret bound above can be easily turned into a uniform root-KL bound under the following additional assumption (the result follows from Lemma 3 in Appendix B).

**Corollary 2.** *If, in addition to the assumptions of Corollary 1, there exist $k_1 \in \mathbb{R}$ and $k_2 \in \mathbb{R}_+$ with $f(x) \leq k_1 + k_2 x \sqrt{\log(1 + x)}$, then*

$$R_T(q) \leq \sqrt{\tfrac{T+1}{c_2}} \left( k_1 + k_2 \sqrt{1 + \mathrm{KL}(q \,\|\, \pi)} \right) + \tfrac{c_1}{\sqrt{c_2}} \sqrt{T}.$$

Under different conditions on $f$, we also derive variance bounds with respect to the (intermediate) predictive distributions, similar to AdaHedge [v+11] and AdaFTRL [OP15b], or with respect to a "prior", as in [A21]. We state the conditions on the regularizer and the corresponding regret bounds in the following corollary.

**Corollary 3.** *Suppose $\nu \in \mathcal{M}^\infty(\Theta, \Sigma)$ and $f : \mathcal{S}(\nu) \to \mathbb{R}$ satisfies:*

1. *$f(1/\nu(\Theta)) \geq 0$;*
2. *$f$ is twice continuously differentiable, $f'' > 0$ on $\mathrm{interior}(\mathcal{S}(\nu))$, and $f''(0+) > 0$.*

*For any sequence $(\ell_t)_{t \in \mathbb{N}} \subseteq \mathcal{L}_{[0,1]}^\infty$, $q \in \mathcal{M}_\nu(\Theta, \Sigma)$, and $T$, $\Psi_f^\nu$-regularized FTRL achieves*

1. *If $1/f''(x) \leq Cx$, $\eta_{t+1} = (C[1/2 + \sum_{s=1}^{t-1} \mathrm{Var}_{\theta \sim \nu^{(\hat{z}_{s+1})}} \ell_s(\theta)])^{-1/2}$ gives*

$$R_T(q) \leq [\Psi_f^\nu(\mathrm{d}q/\mathrm{d}\nu) + 1] \sqrt{C[1/2 + \sum_{t=1}^{T} \mathrm{Var}_{\theta \sim \nu^{(\hat{z}_{t+1})}} \ell_t(\theta)]}.$$

2. *If* $1/f'' \leq C$, $\eta_{t+1} = (C \nu(\Theta)[1/4 + \sum_{s=1}^{t} \mathrm{Var}_{\theta \sim \bar{\nu}} \ell_s(\theta)])^{-1/2}$ *for* $\bar{\nu} = \nu/\nu(\Theta)$ *gives*

$$R_T(q) \leq [\Psi_f^\nu(\mathrm{d}q/\mathrm{d}\nu) + 1]\sqrt{C \nu(\Theta)[1/4 + \sum_{t=1}^{T} \mathrm{Var}_{\theta \sim \bar{\nu}} \ell_t(\theta)]}.$$

### 3.1 Examples of $f$-divergence FTRL

We now apply all three of these corollaries to more concrete choices of $f$. Note that in the setting of Corollary 1, since $\nu$ is a probability distribution and $f$ is convex with $f(1) = 0$, the regularizer corresponds to an $f$-divergence, which we denote by $\Psi_f^\nu(\mathrm{d}q/\mathrm{d}\nu) = D_f(q \| \nu)$ for any $q \in \mathcal{M}_\nu(\Theta, \Sigma)$. In this case, we call $\nu$ a *prior* and refer to the algorithm as *$f$-divergence* FTRL. To demonstrate the utility of Corollary 1, we now show how our result recovers the classical analysis for Hedge and applies to our new root-logarithmic regularizer, both examples of $f$-divergence FTRL. We also use our general expression for the solution to FTRL with linearly decomposable regularizers, given by Lemma 1 in Section 5, to obtain explicit, novel expressions for some solutions of $f$-divergence FTRL.

**Example 1** (Hedge). *The regularizer corresponding to* Hedge *(and the Gibbs posterior from Bayesian inference) is given by* $f(x) = x \log x$, *which satisfies* $D_f(\cdot \| \pi) = \mathrm{KL}(\cdot \| \pi)$ *for any* $\pi \in \mathcal{M}(\Theta, \Sigma)$. *This choice satisfies the assumptions of Corollary 1 with* $c_1 = 1 + 1/e$ *and* $c_2 = 1$, *since* $1/f''$ *is increasing, but* does not *satisfy the conditions of Corollary 2. Thus, we do not obtain uniform root-KL bounds for* Hedge; *we conjecture this is in fact a limitation of* Hedge *and not merely an artifact of our analysis. We can, however, apply Corollary 3 with* $C = 1$ *to obtain*

$$R_T(q) \leq \left[\mathrm{KL}(q \| \nu) + 1\right]\sqrt{1/2 + \sum_{t=1}^{T} \mathrm{Var}_{\theta \sim \nu^{(\hat{z}_{t+1})}} \ell_t(\theta)}.$$

*Further, by Lemma 1,* $f$-divergence *FTRL with regularizer scaling* $\eta_{t+1}$ *recovers the familiar formula*

$$x_{t+1}(\theta) = \frac{\exp(-\eta_{t+1} L_t(\theta))}{\int \exp(-\eta_{t+1} L_t(\vartheta))\nu(\mathrm{d}\vartheta)}. \qquad \triangleleft$$

**Example 2** ($\chi^2$-divergence). *Alquier [A21] analyzed* $f$-divergence *FTRL with constant regularizer scalings, obtaining variance bounds (Theorem 2.1) that require knowledge of the variance to tune the regularizer scaling. He specifically focused on the KL-divergence, covered by the previous example, and the* $\chi^2$-divergence, *corresponding to* $f(x) = x^2 - 1$. *This* $f$ *clearly satisfies the conditions of Corollary 1 with* $c_1 = c_2 = 2$, *so we match the optimized bound of Alquier [A21, Corollary 2.4] with*

$$R_T(q) \leq \sqrt{T} \chi^2(q \| \nu) + \sqrt{2T}.$$

*Further, the conditions of Corollary 3 are satisfied with* $C = 1/2$, *so we obtain the novel, potentially much smaller variance bound (without requiring advance knowledge of the variances)*

$$R_T(q) \leq \tfrac{1}{\sqrt{2}} \left[\chi^2(q \| \nu) + 1\right]\sqrt{1/4 + \sum_{t=1}^{T} \mathrm{Var}_{\theta \sim \nu} \ell_t(\theta)}.$$

*By Lemma 1, the* $\chi^2$-divergence *FTRL solution with regularizer scaling* $\eta_{t+1}$ *is*

$$x_{t+1}(\theta) = \tfrac{1}{2}\left(k_{t+1}^* - \eta_{t+1} L_t(\theta)\right)_+$$

*where* $k_{t+1}^* \in \mathbb{R}$ *solves* $\int[\frac{1}{2}(k_{t+1}^* - \eta_{t+1} L_t(\theta))_+]\nu(\mathrm{d}\theta) = 1$, *which matches the formula obtained by Alquier [A21, Example 3.2].* $\qquad \triangleleft$

**Example 3** (abNormal). *We call* $f$-divergence *FTRL with any* $f$ *satisfying the conditions of Corollary 2* abNormal. *One such example is* $f(x) = \int_1^x \sqrt{2\log(1+s)}\,\mathrm{d}s$, *which satisfies the conditions of Corollary 1 with* $c_1 = 2$ *and* $c_2 = 1/\sqrt{2}$ *since* $f$ *is increasing, and satisfies the conditions of Corollary 2 with* $k_1 = 0$ *and* $k_2 = \sqrt{2}$. *Thus, we obtain*

$$R_T(q) \leq 2\sqrt{(T+1)(1 + \mathrm{KL}(q \| \nu))} + \sqrt{8T}.$$

*The* $f$-divergence *FTRL solution with regularizer scaling* $\eta_{t+1}$ *is given by*

$$x_{t+1}(\theta) = \exp\left\{\left(k_{t+1}^* - \eta_{t+1} L_t(\theta)\right)_+^2/2\right\} - 1,$$

*where* $k_{t+1}^* \in \mathbb{R}$ *solves* $\int[\exp\{(k_{t+1}^* - \eta_{t+1} L_t(\theta))_+^2/2\} - 1]\nu(\mathrm{d}\theta) = 1$. *Note that this formula is heuristically similar to* NormalHedge *when* $|\Theta| = N$, *which assigns weights to experts according to*

$$w_t(i) \propto [\textstyle\sum_{s=1}^{t-1} \langle \ell_s, w_s \rangle - L_{t-1}(i)]_+ \exp\left([\textstyle\sum_{s=1}^{t-1} \langle \ell_s, w_s \rangle - L_{t-1}(i)]_+^2/2c_{t-1}\right),$$

*where* $c_{t-1}$ *solves* $\sum_{i \in [N]} \exp([\sum_{s=1}^{t-1} \langle \ell_s, w_s \rangle - L_{t-1}(i)]_+^2/2c_{t-1}) = eN$. $\qquad \triangleleft$

## 3.2   Lower bound for quantile regret

Bounds for quantile regret and KL regret can be related by observing a) that for $\delta_{(i_\epsilon)}$ denoting a point-mass on the $i_\epsilon$th best expert with respect to $L_T$,

$$R_T(\delta_{(i_\epsilon)}) \leq R_T(u_\epsilon),$$

where $\epsilon = i_\epsilon/N$ and $u_\epsilon = \frac{1}{i_\epsilon} \sum_{j=1}^{i_\epsilon} \delta_{(j)}$ is the uniform distribution over the top $i_\epsilon$ experts, and b) that $\mathrm{KL}\left(u_\epsilon \parallel \mathrm{Unif}([N])\right) = \log(1/\epsilon)$. Thus, KL upper bounds are also upper bounds on quantile regret and quantile regret lower bounds are also lower bounds on certain KL regrets. With this in mind, we provide the first general lower bound for quantile regret on $N$ experts. Our lower bound is matching (up to lower order terms) the leading term in our upper bound for quantile regret achieved by abNormal (Example 3) when $\nu$ is uniform on $[N]$ and $q$ is uniform on only the top quantile of experts, establishing the minimax rate of quantile regret as $\sqrt{T \log(1/\epsilon)}$.

**Theorem 1.** *For all $N \in \mathbb{N}$ there exists a probability distribution $p$ on $[0,1]^N$ such that for any sequence of player predictions $(w_t)_{t \in \mathbb{N}} \subseteq \mathrm{simp}([N])$, $i_\epsilon \in \{1, \ldots, \lfloor N/4 \rfloor\}$, and $T \in \mathbb{N}$,*

$$\mathbb{E}_{\ell_{1:T} \sim p^{\otimes T}} R_T(\delta_{(i_\epsilon)}) \geq \sqrt{(T/2)\left(\log(1/\epsilon) - 2\log 2 + 1/\pi\right)} - \sqrt{2/\pi} - 2\log N - \log 2,$$

*where $\delta_{(i_\epsilon)}$ is the point-mass on the $i_\epsilon$th best expert with respect to $L_T$ and $\epsilon = i_\epsilon/N$.*

## 3.3   Intuition for $f$-divergence FTRL in the KL regret paradigm

The conditions on $f$ in Corollary 1 are essentially the minimal conditions needed for the summation terms in Theorem 2 to cancel with each other regardless of the actual losses. Intuitively, to achieve $f$-divergence regret bounds with $f$-divergence FTRL using the bound of Theorem 2, these terms must cancel so that there is no dependence in the final bound on the regularity of the distributions actually selected by the algorithm. The condition on $f$ in Corollary 2 is essentially the minimal condition needed for Jensen's inequality to imply $D_f\left(\cdot \parallel \cdot\right) \lesssim \sqrt{\mathrm{KL}\left(\cdot \parallel \cdot\right)}$. That it is possible to satisfy both of these conditions simultaneously is the crucial observation that enables our result.

All regularizers to which both Corollaries 1 and 2 apply are essentially equivalent to $f(x) = \int_1^x \sqrt{2\log(1+s)}\mathrm{d}s$, meaning they have the same asymptotic growth rate. To see this, first observe that this $f$ has the minimum amount of curvature needed to satisfy $f'' \cdot (f + c_1) \geq c_2$ since $(x\sqrt{\log x}) \cdot (x\sqrt{\log x})''$ is asymptotically constant. Second, this $f$ has the largest asymptotic growth rate that still satisfies $f(x) \leq k_1 + k_2 x\sqrt{\log(1+x)}$. These two facts together constrain the shape of $f$ to the root-logarithmic choice.

# 4   Semi-adversarial regret bounds

We now turn to a another perspective on prediction with expert advice for which FTRL with linearly decomposable regularizers is optimal. The semi-adversarial paradigm (introduced by [BNR20]) consists of a family of constraints on the adversary's choice of loss distribution, and the goal of the player is to learn as well as possible for the true constraint without having to know the constraint in advance. More precisely, the setting is characterized by an unknown *time-homogeneous convex constraint* on the adversary's choice of loss distribution, which is formally represented by a convex set of probability distributions on $\mathcal{L}_{[0,1]}^\infty$, denoted by $\mathcal{D}$. At each round $t$, the adversary is free to select any distribution from $\mathcal{D}$ to sample $\ell_t$ from. Note that when $\mathcal{D}$ is the set of all probability distributions, the worst-case adversarial setting is recovered, and when $\mathcal{D}$ is a singleton, the stochastic setting is recovered. In this section, we describe a new FTRL algorithm (FTRL-CARL) that achieves minimax optimal expected regret without requiring knowledge of $\mathcal{D}$ in advance.

## 4.1   Adaptive minimax optimality

Minimax regret in the semi-adversarial paradigm is quantified using a few key objects that summarize $\mathcal{D}$. The first is the collection of *effective stochastic gaps*, defined for each expert $i \in [N]$ by

$$\Delta_i = \inf_{p \in \mathcal{D}} \max_{i' \in [N]} \mathbb{E}_{\ell \sim p}\left[\ell(i) - \ell(i')\right].$$

Using these, Bilodeau, Negrea, and Roy [BNR20] define the stochastic gap

$$\Delta_0 = \min \left\{ \Delta_i : \ i \in [N], \Delta_i > 0 \right\}.$$

The second object Bilodeau, Negrea, and Roy [BNR20] define is the set of *effective experts*, which is a subset of $[N]$ defined by

$$\mathcal{I}_0 = \{ i \in [N] : \ \Delta_i = 0 \}.$$

This is the set of all experts who are optimal in expectation for some element of $\mathcal{D}$ (or possibly in the limit along some sequence in $\mathcal{D}$). The number of effective experts is then $N_0 = |\mathcal{I}_0|$.

In this paradigm, the goal is to compete against the best expert. Letting $\delta_i$ denote a point-mass on expert $i$, for notational simplicity we set $R_T \equiv \max_{i \in [N]} R_T(\delta_i)$. Further, although the player does not expect to have knowledge of $\mathcal{D}$ in advance, the goal is to develop methods that do as well as they possibly could have *if they had access to properties of $\mathcal{D}$ in advance*. To characterize this, we say an algorithm (which only has knowledge of $N$ in advance) is *adaptively minimax optimal* if there exists a constant $C$ such that, for all $N$ and $(N_0, \Delta_0)$ pairs, the expected regret of the algorithm is within a factor of $C$ from the minimax regret had the algorithm had access to $(N_0, \Delta_0)$ in advance, for sufficiently large $T$ (where sufficiently large may depend on $N$, $N_0$, and $\Delta_0$). For a precise mathematical formulation of this concept, see Section 3.1 of Bilodeau, Negrea, and Roy [BNR20].

When $N_0 = 1$, Proposition 4 of Mourtada and Gaïffas [MG19] shows that the minimax regret is of order no smaller than $(\log N)/\Delta_0$, and when $N_0 > 1$, Theorem 2 of Bilodeau, Negrea, and Roy [BNR20] shows that the minimax regret is of order no smaller than $\sqrt{T \log N_0}$. Bilodeau, Negrea, and Roy [BNR20] prove that Hedge is *not* adaptively minimax optimal in the semi-adversarial paradigm, and their argument applies to other similar Hedge-based algorithms, such as prod [CMS07], AdaHedge [v+11], and Adapt-ML-Prod [GSv14]. Further, they provide an algorithm that achieves

$$\mathbb{E}R_T \lesssim \sqrt{T \log N_0} + \mathbb{I}[N_0 = 1]\frac{\log N}{\Delta_0} + \mathbb{I}[N_0 > 1]\frac{(\log N)^{3/2}}{\Delta_0}. \tag{3}$$

Since $\mathbb{I}[N_0 > 1](\log N)^{3/2}\Delta_0^{-1}$ is lower order when $N_0 > 1$, this algorithm is adaptively minimax optimal. We now present a new algorithm, FTRL-CARL, which is also adaptively minimax optimal *and* achieves a better regret bound for small $T$.

## 4.2 Semi-adversarial regret bound for FTRL-CARL

Let $h_C : [0, 1] \to \mathbb{R}$ be given by

$$h_C(x) = \begin{cases} x\sqrt{2\log(1/x)} - \sqrt{\frac{\pi}{2}}\,\mathrm{erf}\left(\sqrt{\log(1/x)}\right) + x(N-1)\sqrt{\frac{\pi}{2}} & x \in (0, 1] \\ -\sqrt{\pi/2} & x = 0, \end{cases}$$

set $H_C(w) = \sum_{i \in [N]} h_C(w(i))$, and define FTRL-CARL to be FTRL with $\nu$ defined as counting measure, regularizer $\Psi^\nu_{-h_C}(w) = -H_C(w)$, and regularizer scaling $\eta_t = 2/\sqrt{t}$. Note that this corresponds to FTRL with a linearly decomposable regularizer, and an intuitive explanation for this choice of regularizer can be found in Subsection 4.3. We then have the following regret bound for FTRL-CARL, which removes the term $\mathbb{I}[N_0 > 1](\log N)^{3/2}\Delta_0^{-1}$ from Eq. (3).

**Corollary 4.** *For any time-homogeneous convex constraint $\mathcal{D}$,* FTRL-CARL *achieves: For all $T$,*

$$\mathbb{E}R_T \leq \sqrt{2T \log N},$$

*and if $T > 8(\log N_0)\Delta_0^{-2}$,*

$$\mathbb{E}R_T \leq \sqrt{2T \log N_0} + 25\frac{\log N}{\Delta_0}.$$

Corollary 4 follows from Theorem 3 in Appendix D, which is a more refined regret bound.

## 4.3 Intuition for FTRL-CARL in the semi-adversarial paradigm

The CARL regularizer can be motivated by the following intuition from the Hedge algorithm. For i.i.d. losses, the upper bounds for Hedge [Theorem 2, MG19] are only optimal with regularizer scaling

$\eta_t \gtrsim \sqrt{(\log N)/t}$, and matching lower bounds for the adversarial setting suggest this regularizer scaling constraint is actually necessary for optimal performance. Such a regularizer scaling ensures that the weights of each suboptimal (in expectation) expert decay fast enough. In the semi-adversarial paradigm with more than one effective expert, FTRL-CARE (of [BNR20]) can be interpreted as Hedge with an adaptive regularizer scaling that asymptotically satisfies $\eta_t \gtrsim \sqrt{(\log N_0)/t}$. This smaller regularizer scaling applied to the effective experts is necessary to incur asymptotically $\sqrt{T \log N_0}$ regret. However, since this regularizer scaling is smaller, when there are two or more effective experts the weights assigned to the ineffective experts seemingly are slightly too large.

To rectify this, heuristically, it would be ideal to have expert-specific regularizer scalings of size $\sqrt{(\log N_0)/t}$ for the effective experts and $\sqrt{(\log N)/t}$ for the ineffective experts. Since the effective experts will have weights on the order of $1/N_0$ and the ineffective experts will have weights smaller than $1/N$, the expert-specific regularizer scaling $\eta_t(i) = c\sqrt{\log(1/w_t(i))/t}$ may plausibly achieve the desired behaviour.

The weights of Hedge are defined by the equation $\log(1/w_t(i)) = \eta_t(L_{t-1}(i) + \lambda_t)$, where $\lambda_t$ is chosen to ensure the weights are normalized. Replacing $\eta_t$ with our heuristic yields $\log(1/w_t(i)) = c\sqrt{\log(1/w_t(i))/t}\,(L_{t-1}(i) + \lambda_t)$, which can be rearranged to obtain

$$\sqrt{\log(1/w_t(i))} = \frac{c}{\sqrt{t}}(L_{t-1}(i) + \lambda_t).$$

By Lemma 1, this is exactly the formula for the weights produced with FTRL for the regularizer $-H_C$ and regularizer scaling $\eta_t = c/\sqrt{t}$.

As Theorem 3 shows, our modification to the Hedge algorithm is sufficient to yield semi-adversarial regret bounds with expected regret contribution of size $(\log N)/\Delta_0$ from the ineffective experts, improving on the order of the regret bound for FTRL-CARE when $N_0 > 1$.

## 5   FTRL analysis for general expert spaces

Finally, we analyze the general performance of FTRL with a linearly decomposable regularizer. First, Lemma 1 provides a closed-form expression for the FTRL solution on general spaces. Specifically, it reduces solving the FTRL optimization problem of Eq. (2) to finding a root of a one-dimensional equation (Eq. (4)), rather than solving a complex, possibly infinite dimensional, optimization problem. Second, Theorem 2 provides our fundamental regret bound for FTRL with such regularizers, which we have used to obtain the results in Sections 3 and 4. It generalizes the well-known results involving local norms from the finite dimensional case [AR09; ZS19; O19], and retains an additional summation term (usually uniformly bounded in FTRL analyses) that is crucial both for tight root-KL regret bounds and for tight bounds in the semi-adversarial paradigm. All results in this section are proved in Appendix A. For the remainder of this section, let $\nu \in \mathcal{M}^\infty(\Theta, \Sigma)$ be fixed and arbitrary.

### 5.1   Computing FTRL with linearly decomposable regularizers

For continuously differentiable $f : \mathcal{S}(\nu) \to \mathbb{R}$, let $m_{f'} = \inf_{x \in \mathcal{S}(\nu)} f'(x)$, $M_{f'} = \sup_{x \in \mathcal{S}(\nu)} f'(x)$, and $\tau_{f'}(y) = \max(\min(y, M_{f'}), m_{f'})$. We focus on the setting where $f'$ is strictly increasing, and thus $\tau_{f'}(y)$ truncates its argument to the domain of $[f']^{-1}$.

**Lemma 1.** *Suppose $f : \mathcal{S}(\nu) \to \mathbb{R}$ is twice continuously differentiable with $f'' > 0$ on interior$(\mathcal{S}(\nu))$ and $f''(0+) > 0$. For any $L \in \mathcal{L}^\infty$ and $\eta > 0$,*

$$x^*(\theta) = [f']^{-1}(\tau_{f'}(-\eta L(\theta) + k^*))$$

*satisfies $x^* \in \arg\min_{x \in \mathcal{X}(\nu)}\{\langle L,\ x\rangle_\nu + \eta^{-1}\Psi_f^\nu(x)\}$, where $k^* \in \mathbb{R}$ solves*

$$\int [f']^{-1}(\tau_{f'}(-\eta L(\theta) + k^*))\, \nu(\mathrm{d}\theta) = 1. \tag{4}$$

*Further, this solution is unique up to modification on a set of $\nu$-measure $0$.*

For finite expert classes, solving the normalizing equation provided by Lemma 1 to a given precision is essentially the same difficulty as normalizing the weights for the classical Hedge algorithm. For

example, for FTRL-CARL the range of the normalizing constant scales with $\log N$, and solving this using the bisection method would require only $\mathcal{O}(\log\log N)$ times more computation than normalizing Hedge. For uncountable expert classes, many of the general algorithms for prediction with expert advice from previous work are not easily modified to apply, and the standard analyses of regret bounds rely heavily on a finite $N$. In this setting, solving the FTRL normalizing equation is essentially as difficult as normalizing the posterior distribution for Bayesian inference. That is, often computationally intractable, yet regularly studied for its theoretical properties. Extending approximation techniques for Bayesian inference to approximate the solution to this optimization problem is an interesting question for future work.

## 5.2 Choosing $f$ to obtain specific regret bounds

In order to state our generic decomposition for the regret of FTRL algorithms on abstract spaces, we rely on the definition of the one-dimensional *Bregman divergence*, which is defined for any continuously differentiable, strictly convex $f : \mathbb{R} \to \mathbb{R}$ by

$$B_f(x; y) = f(x) - f(y) - f'(y)(x - y).$$

**Theorem 2.** *Suppose $f : \mathcal{S}(\nu) \to \mathbb{R}$ is twice continuously differentiable with $f'' > 0$ on* interior$(\mathcal{S}(\nu))$ *and $f''(0+) > 0$. For all sequences $(\ell_t)_{t\in\mathbb{N}} \subseteq \mathcal{L}^\infty$ and $(m_t)_{t\in\mathbb{N}} \subseteq \mathbb{R}$, there exist $\hat{z}_{t+1} \in \mathrm{Conv}(x_t, x_{t+1})$ and $\tilde{z}_{t+1} \in \mathrm{Conv}(x_t, \tilde{x}_{t+1})$ such that for all $q \in \mathcal{M}_\nu(\Theta, \Sigma)$, $(\overline{z}_{t+1})_{t\in\mathbb{N}} \in \{(\hat{z}_{t+1})_{t\in\mathbb{N}}, (\tilde{z}_{t+1})_{t\in\mathbb{N}}\}$, and $T$, $\Psi_f^\nu$-regularized FTRL achieves*

$$R_T(q) \leq \frac{1}{\eta_{T+1}} \Psi_f^\nu \left( \frac{dq}{d\nu} \right) - \frac{1}{\eta_1} \min_{x\in\mathcal{X}(\nu)} \Psi_f^\nu(x)$$
$$+ \sum_{t=1}^T \left[ \int \frac{\eta_t}{2} \frac{(\ell_t(\theta) - m_t)^2}{f''(\overline{z}_{t+1}(\theta))} \nu(d\theta) - \left( \frac{1}{\eta_{t+1}} - \frac{1}{\eta_t} \right) \Psi_f^\nu(x_{t+1}) \right], \tag{5}$$

*where*

$$\tilde{x}_{t+1} \in \operatorname*{arg\,min}_{x\in\mathcal{L}_+^1(\nu)} \left\{ \langle \ell_t - m_t \mathbb{1}, \ x \rangle_\nu + \frac{1}{\eta_t} \int B_f(x(\theta); x_t(\theta)) \nu(d\theta) \right\}.$$

*Further, if $\inf_{\theta\in\Theta} \ell_t(\theta) \geq m_t$, then $\tilde{z}_{t+1} \leq x_t$ pointwise.*

Theorem 2 is a general result that can be used to prove regret bounds for a variety of settings under the general FTRL framework we describe, as we have already done in Sections 3 and 4. Furthermore, we highlight that the functional form of Theorem 2 makes it clear how to select the regularizers for both the quantile regret and semi-adversarial paradigms. Specifically, in the former, the terms in summation in Eq. (5) are balanced to cancel, while in the latter, the terms in summation in Eq. (5) are balanced to contribute the same order to the regret. These two cases correspond to the relationship

$$f \cdot f'' = \pm 1. \tag{6}$$

Since the generic bound we obtain in Theorem 2 yields components based upon integrals of both $f$ and $1/f''$, balancing these terms without free tuning parameters requires Eq. (6) to approximately hold. Heuristically, trying to balance these terms with tuning parameters rather than by the choice of $f$ seems to lead to non-adaptive or non-uniform bounds. Hedge provides an example of this for both the semi-adversarial and KL regret cases. In the semi-adversarial paradigm, Bilodeau, Negrea, and Roy [BNR20] showed that Hedge cannot be tuned in a way that is minimax optimal and agnostic to the semi-adversarial constraint that prevails. Similarly, without tuning the regularizer scaling to be dependent on the comparator distribution (equivalently, the quantile of interest), KL (and quantile) regret bounds for Hedge are suboptimal.

## 5.3 Applications of root-KL regret bounds for continuous experts

We now briefly discuss two applications that highlight the immediate benefits of our general analysis (and consequently results of Section 3) applying beyond finite expert spaces.

**Predicting as well as the terminal posterior using $f$-divergence FTRL**    A reasonable choice of distribution to measure regret against is the posterior distribution $\hat{\pi}_T$ after having seen $T$ rounds of data. A consequence of Corollaries 1 and 2 is that, for bounded log-likelihoods, the total loss incurred by making predictions according to $f$-divergence FTRL for suitable $f$ is bounded by the loss incurred by the terminal posterior $\hat{\pi}_T$ plus an excess regret of the order $\sqrt{T \, \mathrm{KL}\left(\hat{\pi}_T \parallel \pi\right)}$. This excess loss is smaller than $T + \mathrm{KL}\left(\hat{\pi}_T \parallel \pi\right)$, which is the best available bound for excess loss when predicting according to the posterior $\hat{\pi}_t$ at each round $t \in [T]$, and smaller than $\sqrt{T} \, \mathrm{KL}\left(\hat{\pi}_T \parallel \pi\right)$, which is the best available bound for the excess loss when predicting with Hedge if it is not tuned with *a priori* knowledge of $\mathrm{KL}\left(\hat{\pi}_T \parallel \pi\right)$. These worse excess loss bounds follow from the analysis contained in Zhang [Z06], although he only compares against the "true" data-generating parameter.

**Model selection using $f$-divergence FTRL**    Extending the interpretation of Orabona and Pál [OP16] by Foster, Krishnamurthy, and Luo [FKL20] to countable union model classes, we can consider an infinite sequence of disjoint finite expert classes (note any nested sequence can be made disjoint), $\Theta_1, \Theta_2, \ldots$, and let $\Theta = \bigcup_{m \geq 1} \Theta_m$. Assigning to $\Theta$ the prior $\pi(\theta) \propto \frac{1}{m^2 |\Theta_m|}$ for each $\theta \in \Theta_m$, we recover the following regret bound for $f$-divergence FTRL with regularizer scaling $\eta_t \propto 1/\sqrt{t}$ from Corollary 2:

$$R_T(\delta_\theta) \leq \mathcal{O}\left(\sqrt{T(\log|\Theta_m| + \log m)}\right) \text{ for all } \theta \in \Theta_m \text{ and all } m \in \mathbb{N}.$$

## 6    Limitations

One limitation of our results, which is common in the online learning literature, is that they only apply to bounded losses. The extension of learning theory to unbounded losses has seen increased interest in recent years [GM20; A21; MVZ21], although it remains a major open problem to achieve guarantees for arbitrary losses. In particular, log loss for many nonparametric learners is not covered by the current unbounded loss literature, and will have important implications in statistical learning and density estimation when resolved.

A second limitation of our work is that finding the implicit normalizing constant—$k^*$ in Lemma 1—is computationally difficult, as was also observed by Alquier [A21]. It is at least as costly as finding the normalizing constant for a Bayesian posterior (or running Hedge), which corresponds to one evaluation of the left-hand side of Eq. (4). For Hedge, where $f(x) = x\log(x)$, one evaluation of that equation is sufficient, but in general finding the root of Eq. (4) to a fixed precision will take a number of evaluations depending on the desired precision and the range of possible values, and so FTRL with a general $f$ will be that marginally more computationally expensive than Hedge. Variational approaches for Bayesian inference, which avoid direct normalization of the posterior, may also be applicable for FTRL with a general $f$, and that line of inquiry may lead to novel, efficient, and high-performance learning algorithms.

A final limitation is that we do not obtain variance bounds together with uniform root-KL bounds, although our approach leads to both separately (Corollary 3 and Corollary 2), so we are optimistic our techniques can lead to such bounds, which would resolve multiple open problems.

## Acknowledgments and Disclosure of Funding

JN is supported by an NSERC Vanier Canada Graduate Scholarship and the Vector Institute. BB is supported by an NSERC Canada Graduate Scholarship and the Vector Institute. FO is partly supported by the National Science Foundation under grants no. 1908111 "AF: Small: Collaborative Research: New Representations for Learning Algorithms and Secure Computation" and no. 2046096 "CAREER: Parameter-free Optimization Algorithms for Machine Learning". DMR is supported in part by an NSERC Discovery Grant, an Ontario Early Researcher Award, and a stipend provided by the Charles Simonyi Endowment. We thank Mufan Li and Mahdi Haghifam for helpful feedback on early drafts.

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
