# A  Proofs for FTRL analysis for general expert spaces

We first prove the formula for the FTRL solution prescribed by Lemma 1, which we recall next.

**Lemma 1.** *Suppose $f : \mathcal{S}(\nu) \to \mathbb{R}$ is twice continuously differentiable with $f'' > 0$ on interior$(\mathcal{S}(\nu))$ and $f''(0+) > 0$. For any $L \in \mathcal{L}^\infty$ and $\eta > 0$,*

$$x^*(\theta) = [f']^{-1}(\tau_{f'}(-\eta L(\theta) + k^*))$$

*satisfies $x^* \in \arg\min_{x \in \mathcal{X}(\nu)}\{\langle L,\ x\rangle_\nu + \eta^{-1}\Psi_f^\nu(x)\}$, where $k^* \in \mathbb{R}$ solves*

$$\int [f']^{-1}\left(\tau_{f'}(-\eta L(\theta) + k^*)\right)\nu(\mathrm{d}\theta) = 1. \tag{4}$$

*Further, this solution is unique up to modification on a set of $\nu$-measure $0$.*

In the proof below, we treat the formula for the optimal solution as an educated guess (inspired by the finite-dimensional case using Lagrange multipliers). The proof then verifies that the proposed solution is well-defined and in fact achieves the optimum. It is also possible to derive the optimal solution directly using a version of the Lagrange multiplier method for convex optimization problems on Banach spaces with cone constraints (see, for example, [L69]). However, to apply such a result, one must still verify the existence of Lagrange multipliers. For example, existence of the Lagrange multiplier for the constraint "$x$ integrates to 1" exactly corresponds to existence of a solution to Eq. (4). Thus it is not significantly more or less laborious to take such an approach over the "guess and check" method we have employed.

*Proof of Lemma 1.* Let $\underline{L} = \inf_{\theta \in \Theta} L(\theta)$ and $\overline{L} = \sup_{\theta \in \Theta} L(\theta)$. Since $f'' > 0$, $f'$ is strictly increasing and restricted to a non-negative domain, so $[f']^{-1}$ exists, is strictly increasing, and is non-negative. Let $a = f'(1/\nu(\Theta))$, and note that since $1/\nu(\Theta) \in \mathcal{S}(\nu)$, $\tau_{f'}(a) = a$. Let $g : \mathbb{R} \to \nu(\Theta) \cdot \mathcal{S}(\nu)$ be given by

$$g(k) = \int [f']^{-1}\left(\tau_{f'}(-\eta L(\theta) + k)\right)\nu(\mathrm{d}\theta).$$

First, note that for all $k$, $g(k) \le \nu(\Theta)\sup_{\theta \in \Theta}[f']^{-1}\left(\tau_{f'}(-\eta L(\theta) + k)\right)$. So, since $L \ge \underline{L}$, $g(k) \le \nu(\Theta)[f']^{-1}\left(\tau_{f'}(-\eta\underline{L} + k)\right)$. Thus,

$$
\begin{aligned}
1 &= \int [f']^{-1}(a)\nu(\mathrm{d}\theta) \\
&\le \int [f']^{-1}(\tau_{f'}(-\eta L(\theta) + \eta\overline{L} + a))\nu(\mathrm{d}\theta) \\
&= g(\eta\overline{L} + a) \\
&\le \nu(\Theta)[f']^{-1}(\tau_{f'}(\eta(\overline{L} - \underline{L}) + a)),
\end{aligned}
\tag{7}
$$

where the second inequality follows since $L \le \overline{L}$. Further, non-negativity of $[f']^{-1}$ gives

$$
\begin{aligned}
0 &\le g(\eta\underline{L} + a) \\
&= \int [f']^{-1}(\tau_{f'}(\eta(\underline{L} - L(\theta)) + a))\nu(\mathrm{d}\theta) \\
&\le \int [f']^{-1}(a)\nu(\mathrm{d}\theta) \\
&= 1.
\end{aligned}
\tag{8}
$$

By Leibniz rule and the inverse function theorem, Eqs. (7) and (8) imply that, for all $k \in \left[\eta\underline{L} + a, \eta\overline{L} + a\right]$,

$$
\begin{aligned}
g'(k) &= \int \frac{\mathbb{I}[M_{f'} \ge -\eta L(\theta) + k \ge m_{f'}]}{f'' \circ [f']^{-1}\left(\tau_{f'}(-\eta L(\theta) + k)\right)}\nu(\mathrm{d}\theta) \\
&\le \frac{\nu(\Theta)}{\inf_{\theta \in \left(0, [f']^{-1}(\tau_{f'}(\eta(\overline{L} - \underline{L}) + a))\right)} f''(\theta)} \\
&< \infty.
\end{aligned}
$$

Hence, $g$ is (Lipschitz) continuous on this interval, and then, by the intermediate value theorem, there exists $k^*$ solving Eq. (4).

Let $F : \mathcal{L}^1_+(\nu) \to \mathbb{R}$ be given by

$$F(x) = \langle L,\ x \rangle_\nu + \frac{1}{\eta} \Psi^\nu_f(x).$$

To avoid notational clutter, for the remainder of the proof all statements involving $\theta$ implicitly hold only $\nu$-a.s. Let $x(\theta) = [f']^{-1}\left(\tau_{f'}(-\eta L(\theta) + k^*)\right)$ and note that the definition of $k^*$ implies that $\int x(\theta)\nu(\mathrm{d}\theta) = 1$. We now argue that $-\eta L + k^* \le M_{f'}$.

If $\underline{\nu} = 0$, then $M_{f'} = \infty$, so trivially $-\eta L + k^* \le M_{f'}$.

If $\underline{\nu} \ne 0$, then $\nu$ is purely atomic and, since $f'$ is increasing, $M_{f'} = f'(1/\underline{\nu})$. By definition of $\tau_{f'}$, $x \le 1/\underline{\nu}$. If $x < 1/\underline{\nu}$, then $\tau_{f'}(-\eta L(\theta) + k^*) < M_{f'}$, so $-\eta L(\theta) + k^* \le M_{f'}$. Otherwise, there exists an atom $A \in \Sigma$ for $\nu$ such that $x(\theta) = 1/\underline{\nu}$ for $\theta \in A$. Since $x \in \mathcal{X}(\nu)$, we must have $x(\theta) = \mathbb{I}[\theta \in A]/\underline{\nu}$; equivalently $x$ is the Radon–Nikodym derivative of a single-atom probability measure completely concentrated on $A$. Without loss of generality, $A = \{\theta :\ x(\theta) = 1/\underline{\nu}\}$ and

$$f'(x(\theta)) = m_{f'} + (M_{f'} - m_{f'})\mathbb{I}[\theta \in A].$$

Then, any $k^*$ that satisfies

$$\begin{cases} -\eta L(\theta) + k^* \ge M_{f'}, & \theta \in A \\ -\eta L(\theta) + k^* \le m_{f'}, & \theta \notin A \end{cases}$$

is valid and gives the same solution. In particular, since there is some $k^*$ satisfying this, then $\tilde{k}^* = M_{f'} + \eta[\nu\text{-}\mathrm{ess\,sup}]_{\theta \in A}L(\theta) \le k^* \le m_{f'} + \eta[\nu\text{-}\mathrm{ess\,inf}]_{\theta \notin A}L(\theta)$, which implies $\tilde{k}^*$ is a valid solution. Finally, we claim that $L$ must be constant on $A$. To see this, suppose otherwise there is some $\tilde{L}$ such that for $\theta \in \tilde{A} \subseteq A$, $L(\theta) < \tilde{L}$, and for $\theta \in A \setminus \tilde{A}$, $L(\theta) \ge \tilde{L}$. Then the inverse images of $(\infty, \tilde{L})$ and $[\tilde{L}, \infty)$ must partition $A$ since $L$ is measurable, but one of them must have $\nu$ measure zero since $A$ is an atom. Thus, it is also true that $\tilde{k}^* = M_{f'} + \eta[\nu\text{-}\mathrm{ess\,inf}]_{\theta \in A}L(\theta)$, so we have argued that, without loss of generality, in all cases $-\eta L + k^* \le M_{f'}$.

With this in mind, consider any $z \in \mathcal{X}(\nu)$. Since $f$ is continuously differentiable and convex,

$$F(z) = F(x) + \langle L,\ z - x \rangle_\nu + \frac{1}{\eta} \int \left[ f(z(\theta)) - f(x(\theta)) \right] \nu(\mathrm{d}\theta)$$

$$\ge F(x) + \langle L,\ z - x \rangle_\nu + \frac{1}{\eta} \int f'(x(\theta)) \left[ z(\theta) - x(\theta) \right] \nu(\mathrm{d}\theta)$$

$$= F(x) + \int_{x(\theta)>0} L(\theta) \left[ z(\theta) - x(\theta) \right] \nu(\mathrm{d}\theta) + \int_{x(\theta)=0} L(\theta) \left[ z(\theta) - x(\theta) \right] \nu(\mathrm{d}\theta)$$

$$+ \frac{1}{\eta} \int_{x(\theta)>0} f'(x(\theta)) \left[ z(\theta) - x(\theta) \right] \nu(\mathrm{d}\theta) + \frac{1}{\eta} \int_{x(\theta)=0} f'(x(\theta)) \left[ z(\theta) - x(\theta) \right] \nu(\mathrm{d}\theta)$$

$$= F(x) + \int_{x(\theta)>0} L(\theta) \left[ z(\theta) - x(\theta) \right] \nu(\mathrm{d}\theta) + \int_{x(\theta)=0} L(\theta) \left[ z(\theta) - x(\theta) \right] \nu(\mathrm{d}\theta)$$

$$+ \frac{1}{\eta} \int_{x(\theta)>0} (-\eta L(\theta) + k^*) \left[ z(\theta) - x(\theta) \right] \nu(\mathrm{d}\theta) + \frac{1}{\eta} \int_{x(\theta)=0} m_{f'} \left[ z(\theta) - x(\theta) \right] \nu(\mathrm{d}\theta)$$

$$= F(x) + \int_{x(\theta)>0} \frac{k^*}{\eta} \left[ z(\theta) - x(\theta) \right] \nu(\mathrm{d}\theta) + \int_{x(\theta)=0} \left( L(\theta) + \frac{m_{f'}}{\eta} \right) \left[ z(\theta) - x(\theta) \right] \nu(\mathrm{d}\theta)$$

$$\ge F(x) + \int_{x(\theta)>0} \frac{k^*}{\eta} \left[ z(\theta) - x(\theta) \right] \nu(\mathrm{d}\theta) + \int_{x(\theta)=0} \frac{k^*}{\eta} \left[ z(\theta) - x(\theta) \right] \nu(\mathrm{d}\theta)$$

$$= F(x),$$

(9)

where in the second last step we have used that if $x(\theta) = 0$ then $-\eta L(\theta) + k^* \le m_{f'}$ and $z(\theta) - x(\theta) \ge 0$. Thus, $x$ is a solution to the FTRL equation. Further, since $f$ is strictly convex, equality can only hold in Eq. (9) if $\nu(x = z) = 1$. $\qquad\square$

To prove Theorem 2, we need the analogue of the finite-dimensional first-order optimality condition. For any $V \subseteq \mathcal{L}^1(\nu)$ and any $F : V \to \mathbb{R}$ and $x \in \mathcal{L}^1(\nu)$, the Gateaux derivative (in the direction of $z \in \mathcal{L}^1(\nu)$) is

$$\delta F[x; z] = \lim_{\alpha \to 0} \frac{F(x + \alpha z) - F(x)}{\alpha}.$$

The following result is straightforward: we include a proof for completeness.

**Lemma 2.** *If $V \subseteq \mathcal{L}^1(\nu)$ is convex and $x = \arg\min_{z \in V} F(z)$, then $\delta F[x; z - x] \geq 0$ for all $z \in V$ where the limit exists.*

*Proof of Lemma 2.* Towards a contradiction, suppose there exists $z \in V$ with $\delta F[x; z - x] < 0$. Define $z_\alpha = \alpha z + (1 - \alpha)x$ for all $\alpha \in [0, 1]$. By definition,

$$\delta F[x; z - x] = \lim_{\alpha \to 0} \frac{F(x + \alpha(z - x)) - F(x)}{\alpha} = \lim_{\alpha \to 0} \frac{F(z_\alpha) - F(x)}{\alpha}.$$

By assumption, this implies that for some $\alpha > 0$, $F(z_\alpha) < F(x)$. However, since $z_\alpha \in V$ for all $\alpha \in [0, 1]$ by convexity, this contradicts the optimality of $x$. $\qquad\square$

Next, define the functional Bregman divergence for any convex $V \subseteq \mathcal{L}^1(\nu)$ and any $F : V \to \mathbb{R}$ and $x, z \in \mathcal{L}^1(\nu)$ by

$$B_F(x; z) = F(x) - F(z) - \delta F[z; x - z].$$

We are now able to prove our generic FTRL bound, Theorem 2, which we recall here for completeness.

**Theorem 2.** *Suppose $f : \mathcal{S}(\nu) \to \mathbb{R}$ is twice continuously differentiable with $f'' > 0$ on interior$(\mathcal{S}(\nu))$ and $f''(0+) > 0$. For all sequences $(\ell_t)_{t \in \mathbb{N}} \subseteq \mathcal{L}^\infty$ and $(m_t)_{t \in \mathbb{N}} \subseteq \mathbb{R}$, there exist $\hat{z}_{t+1} \in \mathrm{Conv}(x_t, x_{t+1})$ and $\tilde{z}_{t+1} \in \mathrm{Conv}(x_t, \tilde{x}_{t+1})$ such that for all $q \in \mathcal{M}_\nu(\Theta, \Sigma)$, $(\overline{z}_{t+1})_{t \in \mathbb{N}} \in \{(\hat{z}_{t+1})_{t \in \mathbb{N}}, (\tilde{z}_{t+1})_{t \in \mathbb{N}}\}$, and $T$, $\Psi_f^\nu$-regularized FTRL achieves*

$$\begin{aligned}
R_T(q) \leq\ & \frac{1}{\eta_{T+1}} \Psi_f^\nu \left( \frac{dq}{d\nu} \right) - \frac{1}{\eta_1} \min_{x \in \mathcal{X}(\nu)} \Psi_f^\nu(x) \\
& + \sum_{t=1}^{T} \left[ \int \frac{\eta_t}{2} \frac{(\ell_t(\theta) - m_t)^2}{f''(\overline{z}_{t+1}(\theta))} \nu(d\theta) - \left( \frac{1}{\eta_{t+1}} - \frac{1}{\eta_t} \right) \Psi_f^\nu(x_{t+1}) \right],
\end{aligned} \tag{5}$$

*where*

$$\tilde{x}_{t+1} \in \arg\min_{x \in \mathcal{L}_+^1(\nu)} \left\{ \langle \ell_t - m_t \mathbb{1},\ x \rangle_\nu + \frac{1}{\eta_t} \int B_f(x(\theta); x_t(\theta)) \nu(d\theta) \right\}.$$

*Further, if $\inf_{\theta \in \Theta} \ell_t(\theta) \geq m_t$, then $\tilde{z}_{t+1} \leq x_t$ pointwise.*

*Proof of Theorem 2.* For all $t$, let

$$F_t(x) = \langle L_{t-1},\ x \rangle_\nu + \frac{1}{\eta_t} \Psi_f^\nu(x).$$

By the trivial extension of Orabona [O19, Lemma 7.1] beyond $\mathbb{R}^d$, we have

$$R_T(u) \leq \frac{1}{\eta_{T+1}} \Psi_f^\nu \left( \frac{dq}{d\nu} \right) - \frac{1}{\eta_1} \min_{x \in \mathcal{X}(\nu)} \Psi_f^\nu(x) + \sum_{t=1}^{T} \left[ F_t(x_t) - F_{t+1}(x_{t+1}) + \langle \ell_t,\ x_t \rangle \right].$$

Then,
$$\begin{aligned}
& F_t(x_t) - F_{t+1}(x_{t+1}) + \langle \ell_t,\ x_t \rangle \\
& = F_t(x_t) - F_t(x_{t+1}) + \langle \ell_t,\ x_t - x_{t+1} \rangle - \left( \frac{1}{\eta_{t+1}} - \frac{1}{\eta_t} \right) \Psi_f^\nu(x_{t+1}).
\end{aligned} \tag{10}$$

By Lemma 2 and the definition of $x_t$,

$$B_{F_t}(x_{t+1}; x_t) \leq F_{t+1}(x_{t+1}) - F_t(x_t).$$

Thus, substituting this into Eq. (10),

$$F_t(x_t) - F_{t+1}(x_{t+1}) + \langle \ell_t,\ x_t \rangle$$
$$\leq -\frac{1}{\eta_t} B_{\Psi_f^\nu(\cdot)}(x_{t+1}; x_t) + \langle \ell_t,\ x_t - x_{t+1} \rangle - \left( \frac{1}{\eta_{t+1}} - \frac{1}{\eta_t} \right) \Psi_f^\nu(x_{t+1}),$$

where we have used linearity of the functional Bregman divergence.

Next, for any $x, z \in \mathcal{X}(\nu)$,

$$\delta \Psi_f^\nu(\cdot)[z; x - z] = \lim_{\alpha \to 0} \int \frac{[f(z(\theta) + \alpha(x(\theta) - z(\theta))) - f(z(\theta))]}{\alpha} \nu(\mathrm{d}\theta). \tag{11}$$

For any $\theta \in \Theta$ and $\alpha \in (0, 1)$, since $f'$ is increasing, by the mean value theorem

$$f' \left( \min \left\{ \inf_{\theta' \in \Theta} z(\theta') + \alpha(x(\theta') - z(\theta')),\ \inf_{\theta' \in \Theta} z(\theta') \right\} \right)$$
$$\leq \frac{[f(z(\theta) + \alpha(x(\theta) - z(\theta))) - f(z(\theta))]}{\alpha} \tag{12}$$
$$\leq f' \left( \max \left\{ \sup_{\theta' \in \Theta} z(\theta') + \alpha(x(\theta') - z(\theta')),\ \sup_{\theta' \in \Theta} z(\theta') \right\} \right).$$

Recall the form of $x_t$ given by Lemma 1. For each round $t$, $\sup_{\theta \in \Theta} \{ -\eta_t L_{t-1}(\theta) + k^* \} < \infty$ so $\sup_{\theta \in \Theta} \max \{ x_t(\theta), x_{t+1}(\theta) \} < \infty$. Thus, since by continuity $f'$ is finite on $\mathcal{S}(\nu)$, for $x = x_{t+1}$ and $z = x_t$ the upper bound of Eq. (12) is finite. If $f'(0+) > -\infty$, the lower bound is also finite. If $f'(0+) = -\infty$, then since $L_{t-1}$ is bounded, $x_t$ is bounded away from 0 uniformly and thus the lower bound of Eq. (12) is still finite. Thus, for any $f$ satisfying the conditions of the lemma, we can apply the bounded convergence theorem to Eq. (11) to obtain

$$B_{\Psi_f^\nu(\cdot)}(x_{t+1}; x_t) = \int B_f(x_{t+1}(\theta); x_t(\theta)) \nu(\mathrm{d}\theta).$$

Thus,

$$F_t(x_t) - F_{t+1}(x_{t+1}) + \langle \ell_t,\ x_t \rangle_\nu$$
$$\leq \int (\ell_t(\theta) - m_t)(x_t(\theta) - x_{t+1}(\theta)) - \frac{1}{\eta_t} B_f(x_{t+1}(\theta); x_t(\theta))$$
$$- \left( \frac{1}{\eta_{t+1}} - \frac{1}{\eta_t} \right) f(x_{t+1}(\theta)) \nu(\mathrm{d}\theta),$$

where we have used that any $m_t$ can be added since $x_t \in \mathcal{X}(\nu)$ for all $t$.

We split the proof now to consider the two choices of intermediate points.

*Intermediate Point A*
By the mean value remainder form of Taylor's theorem there exists $\hat{z}_{t+1}(\theta) \in \mathrm{Conv}(x_t(\theta), x_{t+1}(\theta))$ such that

$$B_f(x_{t+1}(\theta); x_t(\theta)) = \frac{1}{2} f''(\hat{z}_{t+1}(\theta))(x_t(\theta) - x_{t+1}(\theta))^2.$$

That is,

$$F_t(x_t) - F_{t+1}(x_{t+1}) + \langle \ell_t,\ x_t \rangle_\nu$$
$$\leq \int (\ell_t(\theta) - m_t)(x_t(\theta) - x_{t+1}(\theta)) - \frac{1}{2\eta_t} f''(\hat{z}_{t+1}(\theta))(x_t(\theta) - x_{t+1}(\theta))^2 \tag{13}$$
$$- \left( \frac{1}{\eta_{t+1}} - \frac{1}{\eta_t} \right) f(x_{t+1}(\theta)) \nu(\mathrm{d}\theta).$$

Since $f''$ is bounded away from zero, we can apply Fenchel-Young point-wise for each $\theta \in \Theta$ to obtain

$$(\ell_t(\theta) - m_t)(x_t(\theta) - x_{t+1}(\theta)) \leq \frac{\eta_t}{2} \frac{(\ell_t(\theta) - m_t)^2}{f''(\hat{z}_{t+1}(\theta))} + \frac{1}{2\eta_t} f''(\hat{z}_{t+1}(\theta))(x_t(\theta) - x_{t+1}(\theta))^2.$$

Substituting into Eq. (13) and summing over $t$ gives the result.

*Intermediate Point B*
Alternatively, let

$$\tilde{x}_{t+1} \in \underset{x \in \mathcal{L}_+^1(\nu)}{\arg\max}\left\{-\langle \ell_t - m_t \mathbb{1},\, \nu^{(x)}\rangle - \frac{1}{\eta_t}\int B_f(x(\theta); x_t(\theta))\nu(\mathrm{d}\theta)\right\},$$

so

$$F_t(x_t) - F_{t+1}(x_{t+1}) + \langle \ell_t,\, x_t\rangle_\nu$$
$$\leq \int (\ell_t(\theta) - m_t)\,(x_t(\theta) - \tilde{x}_{t+1}(\theta))$$
$$-\frac{1}{\eta_t}B_f(\tilde{x}_{t+1}(\theta); x_t(\theta)) - \left(\frac{1}{\eta_{t+1}} - \frac{1}{\eta_t}\right)f(x_{t+1}(\theta))\nu(\mathrm{d}\theta).$$

Again by the mean value remainder form of Taylor's theorem, there exists $\tilde{z}_{t+1}(\theta) \in \mathrm{Conv}(x_t(\theta), \tilde{x}_{t+1}(\theta))$ such that

$$B_f(\tilde{x}_{t+1}(\theta); x_t(\theta)) = \frac{1}{2}f''(\tilde{z}_{t+1}(\theta))\,(x_t(\theta) - \tilde{x}_{t+1}(\theta))^2.$$

The statement follows from the same point-wise application of Fenchel-Young as for intermediate point A.

Finally, note that for each $t$ and $\theta \in \Theta$, solving the gradient equation and applying a very similar argument to the proof of Lemma 1 gives

$$\tilde{x}_{t+1}(\theta) = [f']^{-1}(\tau_{f'}(-\eta_t(\ell_t(\theta) - m_t) + f'(x_t(\theta)))) \leq x_t(\theta),$$

as long as $\inf_{\theta \in \Theta} \ell_t(\theta) \geq m_t$. $\qquad \square$

# B  Proofs for $f$-divergence FTRL bounds

**Corollary 1.** *Suppose $\nu \in \mathcal{M}(\Theta, \Sigma)$ and $f : \mathbb{R}_+ \to \mathbb{R}$ satisfies:*

1. *$f(1) = 0$;*
2. *$f$ is twice continuously differentiable, $f'' > 0$ on $\mathbb{R}_{++}$, and $f''(0+) > 0$;*
3. *either $f$ or $1/f''$ is increasing on $\mathbb{R}_{++}$;*
4. *there exist $c_1, c_2 > 0$ such that $f'' \cdot (f + c_1) \geq c_2$.*

*For any sequence $(\ell_t)_{t\in\mathbb{N}} \subseteq \mathcal{L}_{[0,1]}^\infty$, $q \in \mathcal{M}_\nu(\Theta, \Sigma)$, and $T$, $\Psi_f^\nu$-regularized FTRL with regularizer scaling $\eta_t = \sqrt{c_2/t}$ achieves*

$$R_T(q) \leq \sqrt{\tfrac{T+1}{c_2}}\,\Psi_f^\nu(\mathrm{d}q/\mathrm{d}\nu) + \tfrac{c_1}{\sqrt{c_2}}\sqrt{T}.$$

*Proof of Corollary 1.* We start by applying Theorem 2 using the intermediate point $\tilde{z}_{t+1}$ and with $m_t \equiv 0$. Since $f'' \cdot (f + c_1) \geq c_2$ and either $1/f''$ or $f$ is increasing, this simplifies to

$$R_T(q) \leq \frac{1}{\eta_{T+1}}\Psi_f^\nu(\mathrm{d}q/\mathrm{d}\nu) + \frac{1}{2c_2}\sum_{t=1}^T \eta_t \int f(x_t(\theta))\,(\ell_t(\theta))^2\,\nu(\mathrm{d}\theta) + \frac{c_1}{2c_2}\sum_{t=1}^T \eta_t \int (\ell_t(\theta))^2\,\nu(\mathrm{d}\theta)$$
$$-\sum_{t=1}^T \left(\frac{1}{\eta_{t+1}} - \frac{1}{\eta_t}\right)\Psi_f^\nu(x_{t+1}).$$

Taking $\eta_t = c/\sqrt{t}$ for some $c > 0$ gives

$$\frac{1}{\eta_{t+1}} - \frac{1}{\eta_t} \geq \frac{1}{2c\sqrt{t+1}}.$$

Finally,

$$\sum_{t=1}^T \frac{1}{\sqrt{t}} \leq 1 + \int_1^T \frac{1}{\sqrt{x}}\mathrm{d}x \leq 2\sqrt{T}.$$

Thus, using that $\ell_t \leq 1$ and $\Psi_f^\nu(x_t) \geq 0$ (by Jensen's and $f(1) = 0$),

$$R_T(q) \leq \frac{\sqrt{T+1}}{c} \Psi_f^\nu(\mathrm{d}q/\mathrm{d}\nu) + \sum_{t=1}^{T} \frac{c}{2c_2\sqrt{t}} \Psi_f^\nu(x_t) + \frac{c_1 c}{c_2}\sqrt{T} - \sum_{t=1}^{T} \frac{1}{2c\sqrt{t+1}} \Psi_f^\nu(x_{t+1})$$

$$\leq \frac{\sqrt{T+1}}{c} \Psi_f^\nu(\mathrm{d}q/\mathrm{d}\nu) + \frac{c_1 c}{c_2}\sqrt{T} + \sum_{t=2}^{T} \left( \frac{c}{2c_2\sqrt{t}} \Psi_f^\nu(x_t) - \frac{1}{2c\sqrt{t}} \Psi_f^\nu(x_t) \right),$$

where the last step uses that $x_1 = \nu$. Taking $c = \sqrt{c_2}$ makes the final summation over $t$ non-positive while minimizing the first term. $\qquad\square$

We also have the following result, which implies Corollary 2.

**Lemma 3.** *If $f : \mathbb{R}_+ \to \mathbb{R}$ and there exists $k_1 \in \mathbb{R}$ and $k_2 \in \mathbb{R}_+$ with $f(r) \leq k_1 + k_2 r \sqrt{\log(1+r)}$ for all $r \in \mathbb{R}_+$, then for all $\mu \in \mathcal{M}_\nu(\Theta, \Sigma)$*

$$D_f\left(\mu \| \nu\right) \leq k_1 + k_2 \sqrt{1 + \mathrm{KL}\left(\mu \| \nu\right)}.$$

*Proof of Lemma 3.*

$$D_f\left(\mu \| \nu\right) = \int f\left(\frac{\mathrm{d}\mu}{\mathrm{d}\nu}(\theta)\right) \nu(\mathrm{d}\theta)$$

$$\leq k_1 + k_2 \int \frac{\mathrm{d}\mu}{\mathrm{d}\nu}(\theta) \sqrt{1 + \log\left(\frac{\mathrm{d}\mu}{\mathrm{d}\nu}(\theta)\right)} \, \nu(\mathrm{d}\theta)$$

$$= k_1 + k_2 \int \sqrt{\log\left(1 + \frac{\mathrm{d}\mu}{\mathrm{d}\nu}(\theta)\right)} \, \mu(\mathrm{d}\theta)$$

$$\leq k_1 + k_2 \sqrt{\int \log\left(1 + \frac{\mathrm{d}\mu}{\mathrm{d}\nu}(\theta)\right) \mu(\mathrm{d}\theta)}$$

$$= k_1 + k_2 \sqrt{\int \frac{\mathrm{d}\mu}{\mathrm{d}\nu}(\theta) \log\left(1 + \frac{\mathrm{d}\mu}{\mathrm{d}\nu}(\theta)\right) \nu(\mathrm{d}\theta)}$$

$$\leq k_1 + k_2 \sqrt{1 + \mathrm{KL}\left(\mu \| \nu\right)}.$$

The first inequality is by assumption, the second is Jensen's, and the third is because $1 + (r \log r) \geq r \log(r + 1)$ for all $r > 0$. $\qquad\square$

Finally, we restate and prove Corollary 3.

**Corollary 3.** *Suppose $\nu \in \mathcal{M}^\infty(\Theta, \Sigma)$ and $f : \mathcal{S}(\nu) \to \mathbb{R}$ satisfies:*

1. *$f(1/\nu(\Theta)) \geq 0$;*
2. *$f$ is twice continuously differentiable, $f'' > 0$ on $\mathrm{interior}(\mathcal{S}(\nu))$, and $f''(0+) > 0$.*

*For any sequence $(\ell_t)_{t \in \mathbb{N}} \subseteq \mathcal{L}_{[0,1]}^\infty$, $q \in \mathcal{M}_\nu(\Theta, \Sigma)$, and $T$, $\Psi_f^\nu$-regularized FTRL achieves*

1. *If $1/f''(x) \leq Cx$, $\eta_{t+1} = (C[1/2 + \sum_{s=1}^{t-1} \mathrm{Var}_{\theta \sim \nu^{(\hat{z}_{s+1})}} \ell_s(\theta)])^{-1/2}$ gives*

$$R_T(q) \leq [\Psi_f^\nu(\mathrm{d}q/\mathrm{d}\nu) + 1]\sqrt{C[1/2 + \sum_{t=1}^{T} \mathrm{Var}_{\theta \sim \nu^{(\hat{z}_{t+1})}} \ell_t(\theta)]}.$$

2. *If $1/f'' \leq C$, $\eta_{t+1} = (C\nu(\Theta)[1/4 + \sum_{s=1}^{t} \mathrm{Var}_{\theta \sim \bar{\nu}} \ell_s(\theta)])^{-1/2}$ for $\bar{\nu} = \nu/\nu(\Theta)$ gives*

$$R_T(q) \leq [\Psi_f^\nu(\mathrm{d}q/\mathrm{d}\nu) + 1]\sqrt{C\nu(\Theta)[1/4 + \sum_{t=1}^{T} \mathrm{Var}_{\theta \sim \bar{\nu}} \ell_t(\theta)]}.$$

*Proof of Corollary 3.* Note that, since $f(1/\nu(\Theta)) \geq 0$, Jensen's inequality implies that for any $x \in \mathcal{X}$

$$\Psi_f^\nu(x) = \int f(x(\theta))\nu(\mathrm{d}\theta) \geq \nu(\Theta)f\left(\frac{1}{\nu(\Theta)} \int x(\theta)\nu(\mathrm{d}\theta)\right) \geq 0.$$

First, suppose $1/f''(x) \le Cx$. Since $\eta_t$ is decreasing, Theorem 2 with intermediate point A implies

$$R_T(q) \le \frac{1}{\eta_{T+1}} \Psi_f^\nu \left( \frac{\mathrm{d}q}{\mathrm{d}\nu} \right) + \sum_{t=1}^T \frac{C\eta_t}{2} \int \hat{z}_{t+1}(\theta) \left( \ell_t(\theta) - m_t \right)^2 \nu(\mathrm{d}\theta).$$

Taking $m_t = \int \ell_t(\theta) \hat{z}_{t+1}(\theta) \nu(\mathrm{d}\theta)$ gives

$$R_T(q) \le \frac{1}{\eta_{T+1}} \Psi_f^\nu \left( \frac{\mathrm{d}q}{\mathrm{d}\nu} \right) + \frac{\sqrt{C}}{2} \sum_{t=1}^T \frac{\mathrm{Var}_{\theta \sim \nu^{(\hat{z}_{t+1})}} \ell_t(\theta)}{\sqrt{1/2 + \sum_{s=1}^{t-2} \mathrm{Var}_{\theta \sim \nu^{(\hat{z}_{s+1})}} \ell_s(\theta)}}$$

$$\le \frac{1}{\eta_{T+1}} \Psi_f^\nu \left( \frac{\mathrm{d}q}{\mathrm{d}\nu} \right) + \frac{\sqrt{C}}{2} \sum_{t=1}^T \frac{\mathrm{Var}_{\theta \sim \nu^{(\hat{z}_{t+1})}} \ell_t(\theta)}{\sqrt{\sum_{s=1}^{t} \mathrm{Var}_{\theta \sim \nu^{(\hat{z}_{s+1})}} \ell_s(\theta)}},$$

where we have used that the variance of a random variable in $[0,1]$ is bounded by $1/4$ (see, e.g., Lemma 8 of [BNR20]). By Lemma 4.13 of Orabona [O19], this gives

$$R_T(q) \le \Psi_f^\nu \left( \frac{\mathrm{d}q}{\mathrm{d}\nu} \right) \sqrt{C \left[ 1/2 + \sum_{t=1}^{T-1} \mathrm{Var}_{\theta \sim \nu^{(\hat{z}_{t+1})}} \ell_t(\theta) \right]} + \sqrt{C \sum_{t=1}^{T} \mathrm{Var}_{\theta \sim \nu^{(\hat{z}_{t+1})}} \ell_t(\theta)}.$$

Next, suppose $1/f''(\theta) \le C$. Since $\eta_t$ is decreasing, Theorem 2 with either intermediate point implies

$$R_T(q) \le \frac{1}{\eta_{T+1}} \Psi_f^\nu \left( \frac{\mathrm{d}q}{\mathrm{d}\nu} \right) + \sum_{t=1}^T \frac{\nu(\Theta) C \eta_t}{2} \int \left( \ell_t(\theta) - m_t \right)^2 \overline{\nu}(\mathrm{d}\theta).$$

Taking $m_t = \int \ell_t(\theta) \overline{\nu}(\theta)$ gives

$$R_T(q) \le \frac{1}{\eta_{T+1}} \Psi_f^\nu \left( \frac{\mathrm{d}q}{\mathrm{d}\nu} \right) + \frac{\sqrt{\nu(\Theta) C}}{2} \sum_{t=1}^T \frac{\mathrm{Var}_{\theta \sim \overline{\nu}} \ell_t(\theta)}{\sqrt{1/4 + \sum_{s=1}^{t-1} \mathrm{Var}_{\theta \sim \overline{\nu}} \ell_s(\theta)}}$$

$$\le \frac{1}{\eta_{T+1}} \Psi_f^\nu \left( \frac{\mathrm{d}q}{\mathrm{d}\nu} \right) + \frac{\sqrt{\nu(\Theta) C}}{2} \sum_{t=1}^T \frac{\mathrm{Var}_{\theta \sim \overline{\nu}} \ell_t(\theta)}{\sqrt{\sum_{s=1}^{t} \mathrm{Var}_{\theta \sim \overline{\nu}} \ell_s(\theta)}}$$

By Lemma 4.13 of Orabona [O19], this gives

$$R_T(q) \le \Psi_f^\nu \left( \frac{\mathrm{d}q}{\mathrm{d}\nu} \right) \sqrt{\nu(\Theta) C \left[ 1/4 + \sum_{t=1}^{T} \mathrm{Var}_{\theta \sim \overline{\nu}} \ell_t(\theta) \right]} + \sqrt{\nu(\Theta) C \sum_{t=1}^{T} \mathrm{Var}_{\theta \sim \overline{\nu}} \ell_t(\theta)}.$$

$\square$

## C  Proof of lower bound for quantile regret

**Theorem 1.** *For all $N \in \mathbb{N}$ there exists a probability distribution $p$ on $[0,1]^N$ such that for any sequence of player predictions $(w_t)_{t \in \mathbb{N}} \subseteq \mathtt{simp}([N])$, $i_\epsilon \in \{1, \ldots, \lfloor N/4 \rfloor\}$, and $T \in \mathbb{N}$,*

$$\mathbb{E}_{\ell_{1:T} \sim p^{\otimes T}} R_T(\delta_{(i_\epsilon)}) \ge \sqrt{(T/2) \left( \log(1/\epsilon) - 2 \log 2 + 1/\pi \right)} - \sqrt{2/\pi} - 2 \log N - \log 2,$$

*where $\delta_{(i_\epsilon)}$ is the point-mass on the $i_\epsilon$th best expert with respect to $L_T$ and $\epsilon = i_\epsilon/N$.*

*Proof of Theorem 1.* Let $\Phi(z) \stackrel{def}{=} \mathbb{P}(Z \le z)$ for $Z \sim \mathrm{Nor}(0,1)$ be the normal cumulative distribution function and $\overline{\Phi}(z) \stackrel{def}{=} 1 - \Phi(z)$ be the normal complementary cumulative distribution function. Let $\ell_t(i) \sim \mathrm{Ber}(1/2)$ for all $t \in [T]$ and $i \in [N]$. For a sequence of real values, $S = (s_j)_{j \in [m]}$, denote its empirical cumulative distribution function by $\widehat{F}_S(x) \stackrel{def}{=} \sum_{j \in [m]} \mathbb{I}[x \le s_j]$. For a non-decreasing *càdlàg* function $F$ (right continuous and left limits exist), let its improper inverse be $F^+(y) = \inf \{x : F(x) \ge y\}$. Let $\tilde{L}_t = L_t - (t/2) \mathbb{1}_N$ be the centred cumulative losses. Clearly,

$$\mathbb{E}_{\ell_{1:T} \sim p^{\otimes T}} R_T(\delta_{(i_\epsilon)}) = -\mathbb{E} \widehat{F}_{\tilde{L}_T}^+(\epsilon).$$

Without loss of generality, we can enrich our probability space so that there exists i.i.d. standard normal random variables $Z_T(i) \overset{\text{iid}}{\sim} \text{Nor}(0,1)$ with $L_T(i) = F^+_{\text{Bin}(T,1/2)} \circ \Phi(Z_T(i))$. From Bretagnolle and Massart [BM89, Lemma 4] this coupling satisfies

$$\left| L_T(i) - T/2 - \frac{\sqrt{T}}{2} Z_T(i) \right| \leq 1 + \frac{Z_T(i)^2}{8},$$

which implies

$$\mathbb{E} \max_{i \in [N]} \left| L_T(i) - T/2 - \frac{\sqrt{T}}{2} Z_T(i) \right| \leq 1 + \frac{1}{8} \mathbb{E} \max_{i \in [N]} Z_T(i)^2.$$

Now,

$$\mathbb{E} \max_{i \in [N]} Z_T(i)^2 \leq \inf_{\lambda \in (0,1/2)} \frac{1}{\lambda} \log \sum_{i \in [N]} \mathbb{E} \exp(\lambda Z_T(i)^2)$$

$$= \inf_{\lambda \in (0,1/2)} \frac{1}{\lambda} \log \frac{N}{\sqrt{1 - 2\lambda}}$$

$$= \inf_{\lambda \in (0,1/2)} \left( \frac{\log N}{\lambda} + \frac{1}{\lambda} \log \frac{1}{\sqrt{1 - 2\lambda}} \right).$$

Overapproximating with $\lambda = 1/4$ gives

$$\mathbb{E} \max_{i \in [N]} Z_T(i)^2 \leq 4 \log N + 2 \log 2.$$

From Ali and Chan [AC65], for $\epsilon > 3/4$

$$-\mathbb{E} \widehat{F}^+_{Z_T}(\epsilon) = \mathbb{E} \widehat{F}^+_{-Z_T}(1 - \epsilon)$$

$$\geq \mathbb{E} \Phi^+ \left( \frac{\lfloor (1 - \epsilon) N \rfloor}{N + 1} \right)$$

$$= \mathbb{E} \overline{\Phi}^{-1} \left( 1 - \frac{\lfloor (1 - \epsilon) N \rfloor}{N + 1} \right).$$

Then, since the coupling is monotone, transformations of the order statistics are equal to the order statistics of the transformation, and hence

$$\left| \widehat{F}^+_{-Z_T}(1 - \epsilon) - \widehat{F}^+_{\left( -\frac{2}{\sqrt{T}}(L_T - T/2) \right)}(1 - \epsilon) \right|$$

$$\leq \max_{i \in [N]} \left| Z_T(i) - \left( \frac{2}{\sqrt{T}} (L_T(i) - T/2) \right) \right|.$$

Thus

$$\mathbb{E} \left| \widehat{F}^+_{-Z_T}(1 - \epsilon) - \widehat{F}^+_{\left( -\frac{2}{\sqrt{T}}(L_T - T/2) \right)}(1 - \epsilon) \right| \leq \frac{4 \log N + 2 \log 2}{\sqrt{T}},$$

so

$$\mathbb{E} \widehat{F}^+_{\left( -\frac{2}{\sqrt{T}}(L_T - T/2) \right)}(1 - \epsilon) \geq \overline{\Phi}^{-1} \left( 1 - \frac{(1 - \epsilon) N}{N + 1} \right) - \frac{4 \log N + 2 \log 2}{\sqrt{T}},$$

which implies

$$-\mathbb{E} \widehat{F}^+_{L_T}(\epsilon) \geq \frac{\sqrt{T}}{2} \overline{\Phi}^{-1} \left( \frac{N}{N + 1} \epsilon + \frac{1}{N + 1} \right) - 2 \log N - \log 2$$

$$\geq \frac{\sqrt{T}}{2} \overline{\Phi}^{-1} (2\epsilon) - 2 \log N - \log 2.$$

Now, applying Lemma 4, for $\epsilon < 1/4$

$$-\mathbb{E} \widehat{F}^+_{L_T}(\epsilon) \geq \sqrt{\frac{T}{2} \left( \log(1/\epsilon) - 2 \log 2 + 1/\pi \right)} - \sqrt{2/\pi} - 2 \log N - \log 2.$$

$$\square$$

**Lemma 4.** *For $x > 0$,*

$$\overline{\Phi}(x) \geq \frac{\exp(1/\pi)\exp(-(x+\sqrt{2/\pi})^2/2)}{2},$$

*and for $0 < y < 1/2$,*

$$\overline{\Phi}^{-1}(y) \geq \sqrt{2\log(1/y) - 2\log 2 + 2/\pi} - \sqrt{2/\pi}.$$

*Proof.* The second bound follows from the first, so we will only prove the first. Let $\phi(x)$ denote $\Phi'(x) = \exp(-x^2/2)/\sqrt{2\pi}$, and

$$h(x) = \log \overline{\Phi}(x) - \log \frac{\exp(1/\pi)\exp(-(x+\sqrt{2/\pi})^2/2)}{2}$$
$$= \log \overline{\Phi}(x) - 1/\pi + \log(2) + (x + \sqrt{2/\pi})^2/2.$$

Then, $h(0) = 0$ and

$$h'(x) = -\phi(x)/\overline{\Phi}(x) + (x + \sqrt{2/\pi}),$$

so $h'(0) = 0$ as well.

Next,

$$h''(x) = x\phi(x)/\overline{\Phi}(x) - \left(\phi(x)/\overline{\Phi}(x)\right)^2 + 1,$$

so $h''(0) = 1 - 2/\pi > 0$.

Suppose towards a contradiction that $h''(x_0) = 0$ for some $x_0 > 0$. Rearranging gives

$$x_0 = \phi(x_0)/\overline{\Phi}(x_0) - \left(\phi(x_0)/\overline{\Phi}(x_0)\right)^{-1},$$

so

$$\phi(x_0)/\overline{\Phi}(x_0) = \frac{\sqrt{4 + x_0^2} + x_0}{2}.$$

However, from Birnbaum [B42][2], for all $x > 0$

$$\phi(x)/\overline{\Phi}(x) < \frac{\sqrt{4 + x^2} + x}{2}.$$

Thus $h''(x) \neq 0$ for all $x > 0$. Since $h''$ is continuous, $h''(0) > 0$, and $h''$ has no zeros on $(0, \infty)$, by the intermediate value theorem $h''(x) > 0$ for all $x > 0$. Finally, since $h(0) = h'(0) = 0$ and $h''(x) > 0$ for all $x > 0$, by standard differential inequalities, $h(x) \geq 0$ for all $x \geq 0$. $\qquad\square$

## D   Proof of semi-adversarial regret bound

To analyze FTRL-CARL, we define two more notions to characterize which stage of learning the algorithm is in. Note that each of these stages occur *implicitly* while running FTRL-CARL, and consequently the user does not need to know any of the quantities associated with $\mathcal{D}$.

For each $i \in [N] \setminus \mathcal{I}_0$, let $T_i = \lceil 8(\log N)/\Delta_i^2 \rceil$ and $T_0 = \max_{i \in [N] \setminus \mathcal{I}_0} T_i$. For each $t \in \mathbb{N}$ let $\mathcal{I}_0^{(t)} = \mathcal{I}_0 \cup \{i \in [N] \setminus \mathcal{I}_0 \text{ s.t. } T_i \geq t\}$ and $N_0^{(t)} = |\mathcal{I}_0^{(t)}|$. For each $j \in \{0, 1, ..., N - N_0 - 1\}$, let $T_{(j)}$ be the decreasing ordered values of $T_i$, so that $T_0 = T_{(0)} \geq T_{(1)} \geq ... \geq T_{(N-N_0-1)}$. Let $T_{(N-N_0)} = 0$. Let $\Delta_{(j)}$ be the corresponding increasing ordered values of $\Delta_i$, so that $\Delta_0 = \Delta_{(0)} \leq \Delta_{(1)} \leq ... \leq \Delta_{(N-N_0-1)}$.

We then have the following regret bound for FTRL-CARL, which simplifies to Corollary 4 by lower bounding all $\Delta_i$ by $\Delta_0$ and observing that $W_{j,N,N_0}$ telescopes in the sum. The more refined bound of Theorem 3 will be significantly tighter for small $T$ when only one $\Delta_i$ is small, although the worst-case adversary will choose all $\Delta_i = \Delta_0$.

---

[2]The stated result has a non-strict inequality, however the proof by Jensen's inequality clearly leads to a strict inequality since the function in the application of Jensen's inequality, $t \mapsto 1/t$ is strictly convex and the distribution to which it is applied is not degenerate, it has density proportional to $\phi'(t)$ for $t > 0$.

**Theorem 3.** *For any time-homogeneous convex constraint $\mathcal{D}$, FTRL-CARL achieves:*
*For all $T$,*

$$\mathbb{E}R_T \leq \sqrt{2T \log N},$$

*and if $T > T_0$,*

$$\mathbb{E}R_T \leq \sqrt{2T \log N_0} + 4(\log N) \sum_{j=0}^{N-N_0-1} W_{j,N,N_0} \frac{1}{\Delta_{(j)}}$$

$$+ \frac{5\sqrt{2}}{N\sqrt{\log N}}(e^{-1/2} + \mathbb{I}[N_0 = 1]) \sum_{i \in [N] \setminus \mathcal{I}_0} \frac{\mathbb{I}[T > T_i]}{\Delta_i} + \sqrt{\log N},$$

*where $W_{j,N,N_0} = \frac{1}{\sqrt{\log N}}\left(\sqrt{\log(N_0 + j + 1)} - \sqrt{\log(N_0 + j)}\right)$.*

*Proof of Theorem 3.* Since $N < \infty$, we wish to apply Lemma 1 and Theorem 2 for $\nu$ equal to counting measure and $\mathcal{S}(\nu) = [0,1]$. It is straightforward to verify that $-h_C''(x) = 1/h_A(x) > 0$ on $(0,1)$. Thus, we can apply Theorem 2 with intermediate point A, so there exists a sequence $v_{t+1} \in \mathrm{Conv}(w_t^{\text{CARL}}, w_{t+1}^{\text{CARL}})$ such that for any sequence $(m_t)_{t \in \mathbb{N}} \subseteq \mathbb{R}$, it holds almost surely that

$$R_T \leq \sqrt{\log N} + \sum_{t=1}^{T} \sum_{i \in [N]} \left[\frac{c}{2\sqrt{t}} h_A(v_{t+1}(i))(\ell_t(i) - m_t)^2 + \frac{1}{c}(\sqrt{t+1} - \sqrt{t})H_C(w_{t+1}^{\text{CARL}}(i))\right],$$

where we have used that $H_C(w) = 0$ for any one-hot $w$, an application of Lemma 9, and the expression for $h''$ again.

Let $h_A(x) = x\sqrt{2\log(1/x)}$ for $x \in (0,1]$ and $h_A(0) = 0$, and set $H_A(w) = \sum_{i \in [N]} h_A(w(i))$. Define $\gamma : \mathrm{simp}([N]) \to \mathrm{simp}([N])$ by

$$\gamma(w) = \begin{cases} (1/N, \ldots, 1/N), & H_A(w) = 0 \\ \left(\frac{h_A(w(i))}{H_A(w)}\right)_{i \in [N]}, & \text{otherwise.} \end{cases}$$

Letting $m_t = \langle \ell_t, \gamma(v_{t+1})\rangle$ then gives

$$R_T \leq \sqrt{\log N} + \sum_{t=1}^{T} \left[\frac{c}{2\sqrt{t}}H_A(v_{t+1}) \operatorname*{Var}_{I \sim \gamma(v_{t+1})}[\ell_t(I)] + \frac{1}{c}(\sqrt{t+1} - \sqrt{t})H_C(w_{t+1}^{\text{CARL}})\right],$$

Since $0 \leq \ell_t \leq 1$, $\operatorname*{Var}_{I \sim \gamma(w)}[\ell_t(I)] \leq 1/4$ for any $w \in \mathrm{simp}([N])$, which combined with $\sqrt{t+1} - \sqrt{t} \leq 1/(2\sqrt{t})$ gives

$$R_T \leq \sqrt{\log N} + \sum_{t=1}^{T} \left[\frac{c}{8\sqrt{t}}H_A(v_{t+1}) + \frac{1}{2c\sqrt{t}}H_C(w_{t+1}^{\text{CARL}})\right]. \tag{14}$$

Applying Lemma 9 to Eq. (14) with any $p$ and $\mathcal{I}_0 = [N]$ gives the worst-case bound of

$$R_T \leq \sqrt{\log N} + \left(\frac{c}{4} + \frac{1}{c}\right) \sum_{t=1}^{T} \frac{1}{2\sqrt{t}}\sqrt{2\log N} \leq \left(\frac{c}{4} + \frac{1}{c}\right)\sqrt{2T \log N}.$$

Next, applying Lemma 9 to Eq. (14) with $p = 1/2$ and $\mathcal{I}_0 = \mathcal{I}_0^{(t)}$,

$$R_T \leq \sqrt{\log N} + \left(\frac{c}{4} + \frac{1}{c}\right) \sum_{t=1}^{T} \frac{1}{2\sqrt{t}}\sqrt{2\log N_0^{(t)}}$$

$$+ \sum_{t=1}^{T} \frac{c}{8\sqrt{t}}\left[\frac{1}{\sqrt{e/2}} \sum_{i \in [N] \setminus \mathcal{I}_0^{(t)}} \sqrt{v_{t+1}(i)} + \mathbb{I}\left[N_0^{(t)} = 1\right]\sqrt{2} \sum_{i \in [N] \setminus \mathcal{I}_0^{(t)}} \sqrt{v_{t+1}(i)}\right]$$

$$+ \sum_{t=1}^{T} \frac{1}{2c\sqrt{t}}\left[\frac{1}{\sqrt{e/2}} \sum_{i \in [N] \setminus \mathcal{I}_0^{(t)}} \sqrt{w_{t+1}^{\text{CARL}}(i)} + \mathbb{I}\left[N_0^{(t)} = 1\right]\sqrt{2} \sum_{i \in [N] \setminus \mathcal{I}_0^{(t)}} \sqrt{w_{t+1}^{\text{CARL}}(i)}\right].$$

Since $v_{t+1} \in \mathrm{Conv}(w_t^{\text{CARL}}, w_{t+1}^{\text{CARL}})$, then $\sqrt{v_{t+1}} \le \sqrt{w_t^{\text{CARL}}} + \sqrt{w_{t+1}^{\text{CARL}}}$, and so

$$
R_T \le \sqrt{\log N} + \left(\frac{c}{4} + \frac{1}{c}\right) \sum_{t=1}^{T} \frac{1}{2\sqrt{t}} \sqrt{2 \log N_0^{(t)}}
$$

$$
+ \sum_{t=1}^{T} \frac{c}{8\sqrt{t}} \left[ \frac{1}{\sqrt{e/2}} \sum_{i \in [N] \setminus \mathcal{I}_0^{(t)}} \sqrt{w_t^{\text{CARL}}(i)} + \mathbb{I}\left[N_0^{(t)} = 1\right] \sqrt{2} \sum_{i \in [N] \setminus \mathcal{I}_0^{(t)}} \sqrt{w_t^{\text{CARL}}(i)} \right]
$$

$$
+ \sum_{t=1}^{T} \left(\frac{c}{8\sqrt{t}} + \frac{1}{2c\sqrt{t}}\right) \left[ \frac{1}{\sqrt{e/2}} \sum_{i \in [N] \setminus \mathcal{I}_0^{(t)}} \sqrt{w_{t+1}^{\text{CARL}}(i)} + \mathbb{I}\left[N_0^{(t)} = 1\right] \sqrt{2} \sum_{i \in [N] \setminus \mathcal{I}_0^{(t)}} \sqrt{w_{t+1}^{\text{CARL}}(i)} \right].
$$

Simplifying gives

$$
R_T \le \sqrt{\log N} + \left(\frac{c}{4} + \frac{1}{c}\right) \sum_{t=1}^{T} \frac{1}{2\sqrt{t}} \sqrt{2 \log N_0^{(t)}}
$$

$$
+ \frac{1}{\sqrt{e/2}} \sum_{i \in [N] \setminus \mathcal{I}_0} \sum_{t=T_i+1}^{T} \frac{1}{2\sqrt{t}} \left[ \frac{c}{4} \sqrt{w_t^{\text{CARL}}(i)} + \left(\frac{1}{c} + \frac{c}{4}\right) \sqrt{w_{t+1}^{\text{CARL}}(i)} \right]
$$

$$
+ \mathbb{I}[N_0 = 1] \sqrt{2} \sum_{i \in [N] \setminus \mathcal{I}_0} \sum_{t=T_0+1}^{T} \frac{1}{2\sqrt{t}} \left[ \frac{c}{4} \sqrt{w_t^{\text{CARL}}(i)} + \left(\frac{1}{c} + \frac{c}{4}\right) \sqrt{w_{t+1}^{\text{CARL}}(i)} \right].
$$

For $T > T_0$, the first term can be decomposed as

$$
\sum_{t=1}^{T} \frac{1}{2\sqrt{t}} \sqrt{2 \log N_0^{(t)}} = \sum_{t=1}^{T_0} \frac{1}{2\sqrt{t}} \sqrt{2 \log N_0^{(t)}} + \sum_{t=T_0+1}^{T} \frac{1}{2\sqrt{t}} \sqrt{2 \log N_0}
$$

$$
\le \sum_{t=1}^{T_0} \frac{1}{2\sqrt{t}} \sqrt{2 \log N_0^{(t)}} + \sqrt{2T \log N_0} - \sqrt{2T_0 \log N_0}.
$$

For the second and third terms, applying Lemma 8, for each $i \in [N] \setminus \mathcal{I}_0$ we have

$$
\mathbb{E} \sum_{t=T_i+1}^{T} \frac{1}{2\sqrt{t}} \left[ \frac{c}{4} \sqrt{w_t^{\text{CARL}}(i)} + \left(\frac{1}{c} + \frac{c}{4}\right) \sqrt{w_{t+1}^{\text{CARL}}(i)} \right]
$$

$$
\le \frac{c}{8} \sum_{t=T_i+1}^{T} \frac{1}{\sqrt{t}} \mathbb{E}\left[(w_t^{\text{CARL}}(i))^p\right] + \left(\frac{1}{2c} + \frac{c}{8}\right) \sum_{t=T_i+2}^{T} \frac{2}{\sqrt{t}} \mathbb{E}\left[(w_t^{\text{CARL}}(i))^p\right]
$$

$$
\le \left(\frac{1}{c} + \frac{3c}{8}\right) \frac{\sqrt{8}(2+c)}{N \Delta_i c \sqrt{\log N}}.
$$

Taking $c = 2$ and combining these results, we have

$$
\mathbb{E}R_T \le \left(\sqrt{2T \log N_0} - \sqrt{2T_0 \log N_0}\right) \mathbb{I}[T > T_0] + \sum_{t=1}^{\min(T,T_0)} \frac{1}{2\sqrt{t}} \sqrt{2 \log N_0^{(t)}}
$$

$$
+ \frac{5\sqrt{2}}{N \sqrt{\log N}} \left(e^{-1/2} + \mathbb{I}[N_0 = 1]\right) \sum_{i \in [N] \setminus \mathcal{I}_0} \frac{\mathbb{I}[T > T_i]}{\Delta_i} + \sqrt{\log N}.
$$

Next, using summation by parts,

$$\sum_{t=1}^{T_0} \frac{1}{2\sqrt{t}} \sqrt{2 \log N_0^{(t)}}$$

$$= \sum_{j=0}^{N-N_0} \sum_{t=T_{(N-N_0-j)}+1}^{T_{(N-N_0-j+1)}} \frac{1}{2\sqrt{t}} \sqrt{2 \log(N-j)}$$

$$\leq \sum_{j=0}^{N-N_0} \left( \sqrt{T_{(N-N_0-j+1)}} - \sqrt{T_{(N-N_0-j)}+1} \right) \sqrt{2 \log(N-j)}$$

$$= \sqrt{2 T_{(1)} \log N_0} - \sqrt{2 T_{(N-N_0)} \log N}$$

$$\quad + \sum_{j=1}^{N-N_0} \sqrt{T_{(N-N_0-j)}} \left( \sqrt{2 \log(N-j+1)} - \sqrt{2 \log(N-j)} \right)$$

$$\leq \sqrt{2 T_0 \log N_0} + \sum_{j=0}^{N-N_0-1} \sqrt{T_{(j)}} \left( \sqrt{2 \log(N_0+j+1)} - \sqrt{2 \log(N_0+j)} \right)$$

$$\leq \sqrt{2 T_0 \log N_0} + 4\sqrt{\log N} \sum_{j=0}^{N-N_0-1} \frac{1}{\Delta_{(j)}} \left( \sqrt{\log(N_0+j+1)} - \sqrt{\log(N_0+j)} \right).$$

In summary, for all $T > 0$

$$\mathbb{E} R_T \leq \sum_{t=1}^{T} \frac{1}{2\sqrt{t}} \sqrt{2 \log N_0^{(t)}} + \sqrt{\log N}$$

$$\quad + \frac{5\sqrt{2}}{N\sqrt{\log N}} \left( e^{-1/2} + \mathbb{I}[N_0 = 1] \right) \sum_{i \in [N] \setminus \mathcal{I}_0} \frac{\mathbb{I}[T > T_i]}{\Delta_i},$$

and for $T > T_0$,

$$\mathbb{E} R_T \leq \sqrt{2T \log N_0} + 4(\log N) \sum_{j=0}^{N-N_0-1} W_{j,N,N_0} \frac{1}{\Delta_{(j)}}$$

$$\quad + \frac{5\sqrt{2}}{N\sqrt{\log N}} \left( e^{-1/2} + \mathbb{I}[N_0 = 1] \right) \sum_{i \in [N] \setminus \mathcal{I}_0} \frac{\mathbb{I}[T > T_i]}{\Delta_i} + \sqrt{\log N},$$

where

$$W_{j,N,N_0} = \frac{\sqrt{\log(N_0+j+1)} - \sqrt{\log(N_0+j)}}{\sqrt{\log N}}.$$

Note that $\sum_{j=0}^{N-N_0-1} W_{j,N,N_0} \leq 1$. $\qquad\square$

### D.1  Bounding the weights of ineffective experts

Let $I_t^* = \arg\min_{i \in [N]} L_t(i)$, so $\delta_{(1)}$ is the one-hot vector with a one on $I_T^*$. Then, we have the following control on the weights.

**Lemma 5.** *For any $i \in [N]$, $w_{t+1}^{\text{CARL}}$ satisfies*

$$w_{t+1}^{\text{CARL}}(i) \leq \exp \left\{ -c^2 \frac{(L_t(i) - L_t(I_t^*))^2}{2(t+1)} \right\}.$$

*Proof of Lemma 5.* The Lagrangian corresponding to the optimization problem defining $w_{t+1}^{\text{CARL}}$ is

$$\mathcal{L}(w; \lambda, \alpha_1, \ldots, \alpha_N) = \langle L_t,\ w \rangle - \frac{\sqrt{t+1}}{c} H_C(w) + \lambda \left( 1 - \langle \mathbf{1},\ w \rangle \right) + \sum_{i \in [N]} \alpha_i w(i).$$

It is straightforward to verify that $h'_C(x) = \sqrt{2\log(1/x)}$, so

$$\frac{\partial}{\partial w(i)}\mathcal{L}(w; \lambda, \alpha_1, \ldots, \alpha_N) = L_t(i) - \frac{\sqrt{t+1}}{c}\sqrt{2\log(1/w(i))} - \lambda + \alpha_i.$$

Setting this to 0 gives

$$\lambda^* = L_t(i) - \frac{\sqrt{t+1}}{c}\sqrt{2\log(1/w(i))} + \alpha_i.$$

Since $H_C$ is invariant to reordering of the expert indices, $w_{t+1}^{\text{CARL}}$ is monotonically non-decreasing in $L_t$, so $w_{t+1}^{\text{CARL}}(I_t^*) \geq 1/N$ and consequently $\alpha_{I_t^*} = 0$. If $w_{t+1}^{\text{CARL}}(i) = 0$, the statement of the lemma trivially holds. Otherwise, $\alpha_i = 0$, so by definition of $\lambda^*$ we have

$$\sqrt{2\log(1/w_{t+1}^{\text{CARL}}(i))} - \sqrt{2\log(1/w_{t+1}^{\text{CARL}}(I_t^*))} = \frac{c}{\sqrt{t+1}}\left(L_t(i) - L_t(I_t^*)\right).$$

The result follows by rearranging this equation and using that $\sqrt{a} - \sqrt{b} \leq \sqrt{a-b}$ for all $a \geq b \geq 0$ and $w_{t+1}^{\text{CARL}}(i) \leq w_{t+1}^{\text{CARL}}(I_t^*)$. $\qquad\square$

Next, we extend Theorem 1 of Bilodeau, Negrea, and Roy [BNR20] to depend on the individual ineffective experts effective gaps.

**Lemma 6.** *For all $N \geq 2$, convex sets $\mathcal{D} \subseteq \mathcal{M}(\hat{\mathcal{Y}}^N \times \mathcal{Y})$, $\lambda > 0$, $t$, and $i \in [N] \setminus \mathcal{I}_0$, if the environment is subject to the time-homogeneous convex constraint then*

$$\mathbb{E}\exp\left\{\lambda\left(L_t(I_t^*) - L_t(i)\right)\right\} \leq \exp\left\{T\left[\lambda^2/2 - \lambda\Delta_i\right]\right\}.$$

*Proof of Lemma 6.* The argument follows nearly identically to the one given by Bilodeau, Negrea, and Roy [BNR20]. However, in the second last step in their argument, note that

$$\inf_{w \in \text{simp}(\mathcal{I}_0)} \mathbb{E}_\ell \exp\left\{\lambda[\langle\ell(\mathcal{I}_0), w\rangle - \ell(i)]\right\} \leq e^{\lambda^2/2 - \lambda\Delta_i}.$$

The last step is then applied in exactly the analogous way. $\qquad\square$

Using these two results, we control the expected size of the weight of an ineffective expert.

**Lemma 7.** *For every $t$, $i \in [N] \setminus \mathcal{I}_0$, and $p > 0$*

$$\mathbb{E}\left[(w_t^{\text{CARL}}(i))^p\right] \leq 2\exp\left\{-t\frac{pc^2}{2(\sqrt{2}+c\sqrt{p})^2}\Delta_i^2\right\}.$$

*Proof of Lemma 7.* For any $s \in (0,1)$,

$$\mathbb{E}\left[(w_t^{\text{CARL}}(i))^p\right] \leq \mathbb{E}\left[(w_t^{\text{CARL}}(i))^p : L_t(i) - L_t(I_t^*) \geq s\Delta_i t\right] + \mathbb{P}\left[L_t(i) - L_t(I_t^*) < s\Delta_i t\right].$$

From Lemma 5,

$$\mathbb{E}\left[(w_t^{\text{CARL}}(i))^p : L_t(i) - L_t(I_t^*) \geq s\Delta_i t\right]$$
$$\leq \mathbb{E}\left[\exp\left\{-pc^2\frac{(L_t(i) - L_t(I_t^*))^2}{2(t+1)}\right\} : L_t(i) - L_t(I_t^*) \geq s\Delta_i t\right]$$
$$\leq \exp\left\{-ps^2c^2\Delta_i^2\frac{t^2}{2(t+1)}\right\}$$
$$\leq \exp\left\{-tps^2c^2\Delta_i^2/4\right\}.$$

Next, using the Cramèr-Chernoff Method and Lemma 6,

$$
\begin{aligned}
&\mathbb{P}\left[L_t(i) - L_t(I_t^*) < s\Delta_i t\right] \\
&= \mathbb{P}\left[L_t(I_t^*) - L_t(i) > -s\Delta_i t\right] \\
&\leq \inf_{\lambda > 0} \frac{\mathbb{E}\exp\left\{\lambda\left(L_t(I_t^*) - L_t(i)\right)\right\}}{\exp\left\{-\lambda s\Delta_i t\right\}} \\
&\leq \inf_{\lambda > 0} \frac{\exp\left\{t\left(\lambda^2/2 - \lambda\Delta_i\right)\right\}}{\exp\left\{-\lambda s\Delta_i t\right\}} \\
&= \inf_{\lambda > 0} \exp\left\{t\left(\lambda^2/2 - (1-s)\lambda\Delta_i\right)\right\} \\
&= \exp\left\{-t\left(1-s\right)^2\Delta_i^2/2\right\},
\end{aligned}
$$

where the last step follows from taking $\lambda = \Delta_i(1-s)$. These two terms decay at the same rate if $ps^2c^2 = 2(1-s)$; that is, $s = \sqrt{2}/(\sqrt{2} + c\sqrt{p})$. Using this value,

$$
\mathbb{E}\left[(w_t^{\text{CARL}}(i))^p\right] \leq 2\exp\left\{-t\frac{pc^2}{2(\sqrt{2} + c\sqrt{p})^2}\Delta_i^2\right\}.
$$

$\square$

This single-round bound can now be summed to compute the overall regret contribution of ineffective experts.

**Lemma 8.** *For every $i \in [N] \setminus \mathcal{I}_0$ and $p > 0$,*

$$
\sum_{t=T_i+1}^{\infty} \frac{1}{\sqrt{t}}\mathbb{E}\left[(w_t^{\text{CARL}}(i))^p\right] \leq \frac{2\sqrt{2}(\sqrt{2} + c\sqrt{p})}{N\Delta_i c\sqrt{p}\log N}.
$$

*Proof of Lemma 8.* Using Lemma 7, we have

$$
\begin{aligned}
\sum_{t=T_i+1}^{\infty} \frac{1}{\sqrt{t}}\mathbb{E}\left[(w_t^{\text{CARL}}(i))^p\right] &\leq 2\sum_{t=T_i+1}^{\infty} \frac{1}{\sqrt{t}}\exp\left\{-t\frac{pc^2}{2(\sqrt{2} + c\sqrt{p})^2}\Delta_i^2\right\} \\
&\leq 2\int_{T_i}^{\infty} \frac{1}{\sqrt{t}}\exp\left\{-t\frac{pc^2}{2(\sqrt{2} + c\sqrt{p})^2}\Delta_i^2\right\}\mathrm{d}t.
\end{aligned}
$$

For any $C > 0$, substituting $r^2/2 = tC$ and using $\mathbb{P}(Z > a) \leq \frac{\exp(-a^2/2)}{a\sqrt{2\pi}}$ for $Z \sim \text{Nor}(0,1)$ gives

$$
\begin{aligned}
\int_{T_i}^{\infty} \frac{1}{\sqrt{t}}\exp\left\{-tC\right\}\mathrm{d}t &= \sqrt{2/C}\int_{\sqrt{2CT_i}}^{\infty} \exp\left\{-r^2/2\right\}\mathrm{d}r \\
&\leq \sqrt{2/C}\,\frac{\exp\left\{-CT_i\right\}}{\sqrt{2CT_i}} \\
&= \frac{\exp\left\{-CT_i\right\}}{C\sqrt{T_i}}.
\end{aligned}
$$

Thus,

$$
\begin{aligned}
\sum_{t=T_i+1}^{\infty} \frac{1}{\sqrt{t}}\mathbb{E}\left[(w_t^{\text{CARL}}(i))^p\right] &\leq 2\frac{\exp\left\{-T_i\frac{pc^2}{2(\sqrt{2}+c\sqrt{p})^2}\Delta_i^2\right\}}{\frac{pc^2}{2(\sqrt{2}+c\sqrt{p})^2}\Delta_i^2\sqrt{T_i}} \\
&\leq \frac{2\sqrt{2}(\sqrt{2} + c\sqrt{p})}{N\Delta_i c\sqrt{p}\log N}.
\end{aligned}
$$

$\square$

## D.2 Entropic concentration of FTRL-CARL regularizer

We now show that the functions used to define the regularizer for $w^{\text{CARL}}$ concentrate like an entropy. This result is the analogue of Lemma 1 of Bilodeau, Negrea, and Roy [BNR20].

**Lemma 9.** *For every $w \in \mathtt{simp}([N])$, $\mathcal{I}_0 \subseteq [N]$ with $|\mathcal{I}_0| = N_0$, and $p \in (0, 1)$,*

$$0 \leq H_C(w) \leq H_A(w) \leq \sqrt{2 \log N_0} + \frac{1}{\sqrt{e(1-p)}} \sum_{i \in [N] \setminus \mathcal{I}_0} w(i)^p + \mathbb{I}[N_0 = 1]\sqrt{2} \sum_{i \in [N] \setminus \mathcal{I}_0} \sqrt{w(i)}.$$

*Proof of Lemma 9.* Using the previously observed expressions for the first and second derivatives, $h_C$ is concave on $(0, 1)$, so $H_C$ is concave on $\mathtt{simp}([N])$. Bauer's maximum principle [B60] says that a concave function on a convex, compact set $S$ attains its minimum at an extreme point of $S$. This means that $H_C$ attains its minimum in the set of extreme points of $\mathtt{simp}([N])$, which corresponds to the set of standard basis vectors. In particular, letting $e_i$ denote the $i$th basis vector,

$$H_C(w) \geq \min_{i \in [N]} H_C(e_i) = 0.$$

For the second inequality, let

$$g(x) = \begin{cases} \sqrt{\frac{\pi}{2}} \operatorname{erf}\left(\sqrt{\log(1/x)}\right) - x(N-1)\sqrt{\frac{\pi}{2}}, & x \in (0, 1] \\ \sqrt{\pi/2}, & x = 0. \end{cases}$$

so that

$$H_C(w) = H_A(w) - \sum_{i \in [N]} g(w(i)).$$

Then,

$$\frac{\mathrm{d}^2}{\mathrm{d}x^2} g(x) = -\frac{1}{2\sqrt{2}x \log^{3/2}(1/x)}.$$

This is negative for $x \in (0, 1)$, so $\sum_{i \in [N]} g(w(i))$ is concave. Again, by Bauer's maximum principle, we have

$$\sum_{i \in [N]} g(w(i)) \geq 0,$$

so

$$H_C(w) \leq H_A(w).$$

For the third inequality, observe that

$$H_A(w) = \sum_{i_0 \in \mathcal{I}_0} w(i_0)\sqrt{2 \log(1/w(i_0))} + \sum_{i \in [N] \setminus \mathcal{I}_0} w(i)\sqrt{2 \log(1/w(i))}.$$

Consider the convex optimization problem

$$\min_{\substack{v \in \mathbb{R}_+^{N_0} \\ \langle \mathbf{1}, v \rangle \leq 1}} \left\{ -\sum_{i_0 \in \mathcal{I}_0} v(i_0)\sqrt{2 \log(1/v(i_0))} \right\}.$$

This has the Lagrangian

$$\mathcal{L}(v; \alpha, \lambda_1, \ldots, \lambda_{N_0}) = -\sum_{i_0 \in \mathcal{I}_0} v(i_0)\sqrt{2 \log(1/v(i_0))} + \alpha(\langle \mathbf{1}, v \rangle - 1) + \sum_{i_0 \in \mathcal{I}_0} \lambda_{i_0} v(i_0).$$

The partial derivatives for $i_0 \in \mathcal{I}_0$ are

$$\partial_{i_0} \mathcal{L}(v; \alpha, \lambda_1, \ldots, \lambda_{N_0}) = -\sqrt{2 \log(1/w(i_0))} + \frac{1}{\sqrt{2 \log(1/w(i_0))}} + \alpha + \lambda_{i_0}.$$

Note that this is undefined at $w(i_0) = 0$, so $\lambda_{i_0} = 0$ for all $i_0$. If $\alpha = 0$, then for each $i_0 \in \mathcal{I}_0$, $w(i_0) = e^{-1/2}$, and this is only feasible when $N_0 = 1$. In this case,

$$\sum_{i_0 \in \mathcal{I}_0} w(i_0)\sqrt{2 \log(1/w(i_0))} = \frac{1}{\sqrt{e}}.$$

Otherwise, $\alpha > 0$, and since $-\sqrt{2\log(1/x)} + \frac{1}{\sqrt{2\log(1/x)}}$ is monotonic in $x$, all the $w(i_0)$ are equal. By the K.K.T. condition, this implies $w(i_0) = 1/N_0$ for each $i_0$, so

$$\sum_{i_0 \in \mathcal{I}_0} w(i_0)\sqrt{2\log(1/w(i_0))} = \sqrt{2\log N_0}.$$

That is, if $N_0 \geq 2$,

$$H_A(w) \leq \sqrt{2\log N_0} + \sum_{i \in [N]\setminus\mathcal{I}_0} w(i)\sqrt{2\log(1/w(i))}. \tag{15}$$

Next, if $\mathcal{I}_0 = \{i_0\}$, then we have

$$
\begin{aligned}
H_A(w) &= w(i_0)\sqrt{2\log(1/w(i_0))} + \sum_{i \in [N]\setminus\mathcal{I}_0} w(i)\sqrt{2\log(1/w(i))} \\
&\leq \sqrt{2w(i_0)\log(1/w(i_0))} + \sum_{i \in [N]\setminus\mathcal{I}_0} w(i)\sqrt{2\log(1/w(i))} \\
&\leq \sqrt{2(1 - w(i_0))} + \sum_{i \in [N]\setminus\mathcal{I}_0} w(i)\sqrt{2\log(1/w(i))} \\
&= \sqrt{2\sum_{i \in [N]\setminus\mathcal{I}_0} w(i)} + \sum_{i \in [N]\setminus\mathcal{I}_0} w(i)\sqrt{2\log(1/w(i))} \\
&\leq \sum_{i \in [N]\setminus\mathcal{I}_0} \left[ \sqrt{2w(i)} + w(i)\sqrt{2\log(1/w(i))} \right],
\end{aligned}
\tag{16}
$$

where we have used $x\log(1/x) \leq 1 - x$ for all $x > 0$ and that $\sqrt{x + y} \leq \sqrt{x} + \sqrt{y}$ for all $x, y > 0$.

Finally, for any $p \in (0, 1)$, let $f(x) = x^{1-p}\sqrt{2\log(1/x)}$. Observe that $f(0^+) = 0$, $f(1) = 0$, and

$$f'(x) = (1 - p)x^{-p}\sqrt{2\log(1/x)} - \frac{x^{-p}}{\sqrt{2\log(1/x)}}.$$

Thus, the only critical point of $f$ occurs at $x_0 = e^{-\frac{1}{2(1-p)}}$. Substituting this gives

$$f(x) \leq \frac{1}{\sqrt{e(1 - p)}}.$$

It follows that for $x \in [0, 1]$

$$x\sqrt{2\log(1/x)} \leq \frac{x^p}{\sqrt{e(1 - p)}} \tag{17}$$

Combining Eqs. (15) to (17) gives the third inequality. □

# E  Experiments

In this section, we provide an empirical evaluation of our proposed algorithms and compare them to existing algorithms in the respective paradigms. In particular, for the quantile regret paradigm we implement the version of abNormal proposed in Example 3, while for the semi-adversarial paradigm we implement FTRL-CARL and compare it to Meta-CARE [BNR20] and Hedge with a decreasing regularizer scaling [MG19]. In both cases we generate synthetic data, with specific details in the respective sections. All algorithms use standard tuning for hyperparameters; the specific details can be found in the code, available at https://github.com/blairbilodeau/neurips-2021.

## E.1  Quantile regret

For this experiment, we follow the same setup as Chaudhuri, Freund, and Hsu [CFH09]. The losses are generated from the *Hadamard* matrix of dimension 64, where the row with constant values is removed, the remaining rows are duplicated with their signs inverted, and all the rows are repeated

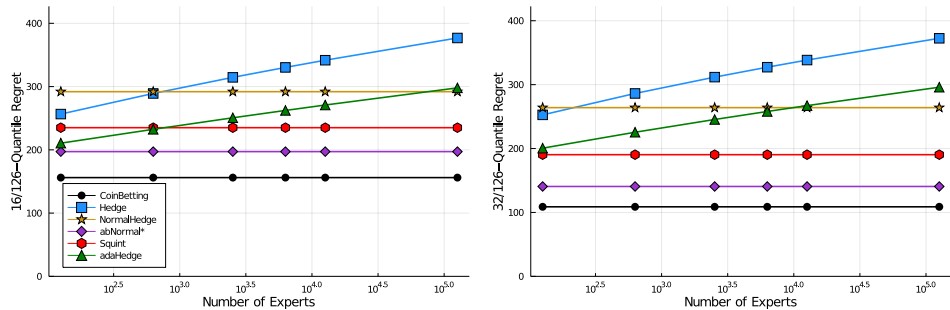

Figure 1: Quantile regret to the top $K/n$ proportion of experts versus number of experts $N$.

horizontally in order to get $T = 32768$ columns and $n = 126$ rows. Each row $i$ represents the losses for expert $i$ from $t = 1, \ldots, T$. Next, given a parameter $K$, which represents the number of *good* experts (out of $n$), the value 0.025 is subtracted from each entry in the first $K$ rows. Each row is then duplicated $N/n$ times in order to have $N$ total experts. Finally, the values in the matrix are shifted and normalized in order to be bounded in the range $[0, 1]$. For more details, see Chaudhuri, Freund, and Hsu [CFH09, Section 3]. As noted by Chaudhuri, Freund, and Hsu [CFH09], the replication factor affects the behaviour of algorithms that are tuned in terms of the total number of experts $N$, such as Hedge with a decreasing regularizer scaling [MG19] or AdaHedge [d+14]. On the other hand, parameter-free algorithms such as Squint [Kv15], NormalHedge [CFH09] or CoinBetting [OP16] are not affected by the replication process. The results illustrated in Fig. 1 show that abNormal follows the same behaviour as parameter-free algorithms. In particular, we plot the cumulative regret after $T = 32768$ time-steps against the best expert. Note that the cumulative loss is equal for the $K \times N/n$ *good* experts, so we are effectively plotting the regret against the top $\varepsilon = K/n$-quantile, which is kept constant despite the total number of experts $N$ increasing. We adopt an initial uniform distribution over experts for all the algorithms.

## E.2  Semi-adversarial losses

For this experiment, we compare Hedge with decreasing regularizer scaling $\eta_t \propto 1/\sqrt{t}$, Meta-CARE [BNR20] and FTRL-CARL from Section 4. In this case, we keep the number of total experts fixed to $N = 1000$, and use deterministic losses. When $N_0 = 1$, the best expert has loss 0.4 and the rest have loss 0.5 on every round. When $N_0 = 2$, expert 1 alternates between loss of 0 and loss of 1, expert 2 alternates (on opposite rounds from expert 1) between loss of 1 and loss of 0, and the rest have loss 0.6 on every round. Thus, both $N_0 = 1$ and $N_0 = 2$ correspond to $\Delta_0 = 0.1$. When $N_0 = N$, the first half and second half of the experts alternate between loss 1 and loss 0 on opposite rounds.

As the theory of Bilodeau, Negrea, and Roy [BNR20] prescribes, Hedge does well for $N_0 = 1$ but does equally poorly for $N_0 = 2$ as $N_0 = N$, while Meta-CARE improves on Hedge for $1 < N_0 < N$. As Theorem 3 prescribes, FTRL-CARL does well in all settings, improving on Meta-CARE by adapting faster in the $N_0 = 2$ case.

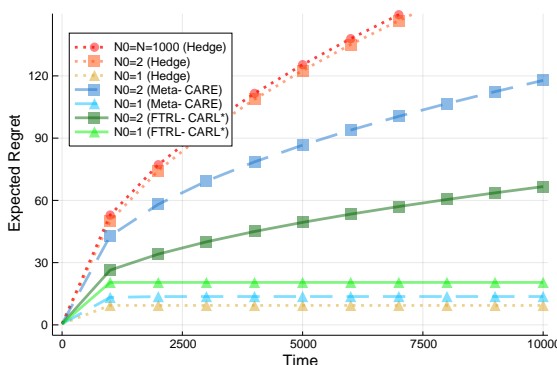

Figure 2: Semi-adversarial paradigm: Expected regret versus time for various values of $N_0$.