# OpenReview forum: "Minimax Optimal Quantile and Semi-Adversarial Regret via Root-Logarithmic Regularizers"
_NeurIPS.cc/2021/Conference — NeurIPS 2021 Poster_

### Official Review · Reviewer_FaPw · 2021-06-25

**Rating:** 7
**Confidence:** 4

**Summary:**

This paper studies the classic expert problem and focuses on developing
parameter-free algorithms with two specific guarantees: 1) better regret
bounds for the semi-adversarial paradigm proposed by Bilodeau et al. 20;
2) a KL-type regret bound similar to that achieved by NormalHedge
(Chaudhuri et al. 09) and other variants, possibly for an uncountable
set of experts. The authors achieve these two goals using the standard
FTRL framework with two new root-logarithmic regularizers, inspired by
their general analysis that requires some specific properties of the
regularizer. A matching lower bound for the KL-type regret is also
provided.

**Limitations And Societal Impact:**

N/A. This is mostly a theoretical work.

**Main Review:**

Overall this paper makes solid contributions to a fundamental problem,
and I support acceptance. What I particularly like is achieving the
KL-type regret with an FTRL approach, different from all existing
algorithms (NormalHedge, AdaNormalHege, Squint, Coin-betting, etc.). I
believe at least several researchers have tried to do so but failed.

However, I do have some concerns and questions:

1) In the semi-adversarial case, what happens when \Delta_0 is very tiny
(or \Delta_i is simply 0 for every i)? Does FTRL-CARL insure some
"safety" guarantee (ideally \sqrt{T\ln N}) always, besides the guarantee
stated in Thm 2?

2) Although achieving KL-type regret with yet another algorithm is nice,
the guarantee in Corollary 2 is in fact strictly worse than that of
AdaNormalHedge (Luo and Schapire 15) and Squint (Koolen and van Erven
15), in the sense that the latter not only achieve the KL bound, but
also some data-dependent bounds which have some nice implications. In
fact, one implication is exactly an logarithmic bound for the stochastic
setting or more generally logarithmic bound plus \sqrt{C} regret for a
corrupted stochastic setting with C being the amount of corruption (this
last result is not stated formally in these papers but the derivation is
straightforward; one can see some examples in the bandit setting such as
[1]). In fact, I personally think the corrupted setting is more natural
than the semi-adversarial setting considered in this work as an
intermediate setting between the stochastic case and the adversarial
case. From this perspective, AdaNormalHedge and Squint appear to be much
more appealing. In fact, there are also recent improvements over these
works as well, such as [2].


On writing:
The paper is generally well-written, but I have some comments below.

1) While I appreciate presenting the result in the most general form
using measure theory language, this also makes the results hard to
interpret in most places. I would prefer presenting the finite case in
the main text and deferring the general case to the appendix. The
semi-adversarial results also seem to be specific to the finite case
anyway? (BTW if this is true it should be mentioned at the beginning of
Sec 4.1.)

2) More explanations after some of the theorems/lemmas (such as Thm 1)
would be very helpful.

3) I wonder whether the proposed algorithms are really that similar to
NormalHedge (which the author claim). Looking at their update rules, I
would only say that they are somewhat connected (maybe I am just not
seeing the deeper connections).


[1] Julian Zimmert and Yevgeny Seldin. Tsallis-inf: An optimal algorithm
for stochastic and adversarial bandits. JMLR 2021.

[2] Liyu Chen, Haipeng Luo, and Chen-Yu Wei. Impossible Tuning Made
Possible: A New Expert Algorithm and Its Applications. COLT 2021.

**Time Spent Reviewing:**

2.5 hours

---

> ### Author Response · Authors · 2021-08-10
> **Response to reviewer FaPw**
>
> We thank the reviewer for their time and detailed comments.
>
> We particularly appreciate that the reviewer has noted the following:
>
> > Overall this paper makes solid contributions to a fundamental problem, and I support acceptance. What I particularly like is achieving the KL-type regret with an FTRL approach, different from all existing algorithms (NormalHedge, AdaNormalHege, Squint, Coin-betting, etc.). I believe at least several researchers have tried to do so but failed.
>
> We strongly agree that this innovation is important for the further development of online learning algorithms, and highlight this point for the other reviewers.
>
> ## Summary
>
> The reviewer’s concerns can be summarized as a) questioning the relevance of the semi-adversarial paradigm versus the corrupted expert paradigm; b) questioning the necessity of the full generality of our results; and c) minor technical questions about our results. We address all of these in detail in what follows; in particular, we a) argue that the semi-adversarial algorithms can be reasonably extended to the corrupted setting, although this is not in the paper; b) justify that the generality of our results for uncountable experts is a significant contribution and is needed for connections to existing work that we make in the main body; and c) answer the technical questions.
>
> ## Addressing: *“I personally think the corrupted setting is more natural than the semi-adversarial setting considered in this work as an intermediate setting between the stochastic case and the adversarial case.”*
>
> FTRL-CARL (and Meta-CARE of [BNR20]) can both be shown to perform well in the corrupted stochastic setting (or the corrupted semi-adversarial setting) as well; the only real modification to the proofs for these cases is that the concentration inequalities need to be modified to account for delay in concentration due to the corruption. Corruption and the semi-adversarial paradigms are not mutually exclusive forms of interpolation between stochastic and adversarial. In particular, using FTRL makes an analysis of corruption simpler, as in [AAKML20] and [ZS21]. In [AAKML20], [LML18], [GKT19] and [ZS21], for large enough T, the modification to the regret bound due to corruption is an additive O(C) term. This can be accounted for in our analysis by simply adapting the definition of $T_0$ to account for corruption relative to the size of the gaps, in both the stochastic and semi-adversarial cases.
>
> ## Addressing: *“The semi-adversarial results also seem to be specific to the finite case anyway?”*
>
> The reviewer is correct that the semi-adversarial paradigm currently only applies to the finite expert setting, and we will make this more explicit in the final version. It remains an interesting open problem to generalize this paradigm appropriately to a continuum of experts.
>
> ## Addressing: *“While I appreciate presenting the result in the most general form using measure theory language, this also makes the results hard to interpret in most places.”*
>
> To handle continuous classes of experts, it is necessary to use Radon-Nikodym derivatives to characterize the “weight vector”. A natural motivating example, which we have not discussed extensively in the paper due to space constraints, is the setting of misspecified Bayesian inference. Recent work [Gv17, GM20] has demonstrated that optimal Bayesian prediction with a misspecified likelihood requires using a tempered likelihood, which has been well-studied (e.g., [Zha06]). Bayesian inference corresponds exactly to FTRL in the continuous case with the Shannon entropy, and the learning rate in FTRL exactly corresponds to the tempering parameter (i.e., the learning rate must be strictly smaller than one under misspecification). Our techniques provide a method to study these techniques even without the IID assumption on the data.
>
> An extension of this idea is to move beyond the Shannon entropy for generalized Bayes procedures, as discussed by [Alq21]. Our analysis furthers this extension, providing novel regret bounds. For instance, in Section 4.2.5, we provide regret guarantees for competing against the terminal posterior distribution in advance when we predict using the root-log regularizer; this is a novel contribution to obtain such results without knowledge of the terminal posterior to tune the learning rate. Finally, even in the discrete setting, to obtain KL-regret guarantees with non-uniform priors it is necessary to use likelihood ratios, which we note required nearly as much effort as rigorously handling the continuous case.
>
> Finally, the generality of Theorem 1 (especially expressing it in terms of Radon-Nikodym derivatives or likelihood ratios) makes it clear how to select the regularizer to achieve sqrt(KL) regret bounds using FTRL, and without such a result it was unclear that it was even possible to construct FTRL regularizers for KL regret bounds. We suspect that this is why achieving such bounds with FTRL had eluded the researchers in the past, as the reviewer noted.
>
> ## Addressing: *“I would prefer presenting the finite case in the main text and deferring the general case to the appendix.”*
>
> Another reviewer has suggested moving the examples earlier in the paper, which will make the direct application of Theorem 1 to the finite case more immediate. However, Theorem 1 is a general result that we expect can be used to prove regret bounds for a variety of settings under the general FTRL framework we describe. We argue its placement in the main body (rather than the appendix) is justified precisely because it is the starting point for the multiple specific regret bounds we provide in the paper. Further, the connection with [Alq21] and to misspecified Bayesian prediction relies heavily on the generality of our results applying to the uncountable setting.
>
> ## Addressing: *“In the semi-adversarial case, what happens when \Delta_0 is very tiny (or \Delta_i is simply 0 for every i)?”*
>
> Since Theorem 2 is stated in terms of individual expert gaps, when $\Delta_0$ tends to zero an extra effective expert is added, and thus the upper bound smoothly transitions to the case when there are $N_0+1$ effective experts. In the worst-case, when all the $\Delta_i$’s go to zero, $T_0$ from Theorem 2 tends to infinity, and thus the upper bound becomes the same as the usual worst-case adversarial upper bound. This behaviour is tight, since the worst case when $\Delta_0$ goes to zero is that all the experts are the same, which is the setting that achieves the $\sqrt{T \log N}$ lower bound (e.g., IID Bernoulli experts suffice).
>
> ## Addressing: *“I wonder whether the proposed algorithms are really that similar to NormalHedge (which the authors claim).”*
>
> The similarity to NormalHedge is indeed heuristic rather than precise. Notably, multiplicative normalization (which NormalHedge uses) can only be achieved by FTRL when using the Shannon entropy as the regularizer, which results in Hedge weights. Consequently, we only advocate that abNormal can be viewed as the appropriate analogue of NormalHedge.
>
> To specify our heuristic justification, consider that when not constrained to the simplex, additive potential functions and additive regularizers are convex conjugates of each other. The various versions of NormalHedge use additive potentials that would have Legendre duals satisfying the essential differential inequality underlying our methods $((f+c_1) \cdot f" \geq c_2)$. For example, the potential corresponding to abNormal in our paper under this relationship would be approximately $\text{Erfi}[x]-x$, where Erfi is the imaginary error function (antiderivative of $\exp(x^2)$). Since the potential given by $\text{Erfi}[x]-x$ and that given by $\exp(x^2/2)$ differ only in lower order terms ($\text{Erfi}[x] ~~ x \exp(x^2)$), we feel justified in saying that abNormal is the closest instantiation of FTRL thus far to match NormalHedge, although we will make it more clear that they are not the same algorithm.
>
> Ignoring the normalization limitation mentioned above, the exact potential from [CFH09] would give a regularizer based on a polylogarithm function that our Theorem 3 could be applied to, obtaining a sqrt(KL) bound. However, due to the normalization this would still not be exactly NormalHedge, and it would be more complicated arithmetically to work with; since we are using FTRL, it is more convenient to design abNormal with the regularizer first rather than the potential.
>
>
> ## References
>
> [Alq21] Alquier, P. (2021). “Non-exponentially weighted aggregation: regret bounds for unbounded loss functions”. Proceedings of the 38th International Conference on Machine Learning.
>
> [AAKML20] Amir, I., Attias, I., Koren, T., Mansour, Y., & Livni, R. (2020). Prediction with Corrupted Expert Advice. In Advances in Neural Information Processing Systems 33.
>
> [BNR20] Bilodeau, B., Negrea, J., & Roy, D. M. (2020). Relaxing the IID Assumption: Adaptively Minimax Optimal Regret via Root-Entropic Regularization. arXiv preprint arXiv:2007.06552.
>
> [CFH09] Chaudhuri, K., Freund, Y., & Hsu, D. J. (2009). A parameter-free hedging algorithm. In Advances in Neural Information Processing Systems 23.
>
> [GKT19] Gupta, A., Koren, T., & Talwar, K. (2019). Better algorithms for stochastic bandits with adversarial corruptions. In Proceedings of the 32nd Conference on Learning Theory.
>
> [LML18] Lykouris, T., Mirrokni, V., & Paes Leme, R. (2018). Stochastic bandits robust to adversarial corruptions. In Proceedings of the 50th Annual ACM SIGACT Symposium on Theory of Computing.
>
> [ZS21] Zimmert, J. & Seldin, Y. (2021). Tsallis-INF: An Optimal Algorithm for Stochastic and Adversarial Bandits. Journal of Machine Learning Research, 22(28), 1-49.
>
> [Zha06] Zhang, T. (2006). “Information-Theoretic Upper and Lower Bounds for Statistical Estimation”. IEEE Transactions on Information Theory, 52, 1307-1321.

---

> > ### Comment · Reviewer_FaPw · 2021-08-19
> > **Follow-up questions**
> >
> > Thanks for the responses to my questions/concerns!
> >
> > I do want to follow up on my comment on the bound of Thm 2 though ($\sqrt{T\ln N_0} + \frac{\ln N}{\Delta_0}$). I don't think this bound transits "smoothly" as you claim in the rebuttal when one particular $\Delta_i$ goes to zero -- in this process the second term first approaches infinity and then suddenly drops to something small when expert $i$ becomes an effective expert, which seems quite weird.
> >
> > My question on "safety guarantee" is also not addressed. I was just asking whether the algorithm guarantees $\sqrt{T\ln N}$ *always*, not when all $\Delta_i$'s are zero. And I am asking this because this worst-case bound could be better than $\sqrt{T\ln N_0} + \frac{\ln N}{\Delta_0}$ sometimes, so ideally one would like to achieve the minimum of them. Could you comment on that?

---

> > > ### Author Response · Authors · 2021-08-19
> > > **Response to follow-up questions from Reviewer FaPw**
> > >
> > > ### Addressing the second point first
> > >
> > > The first part of Theorem 2, $R_T\leq \sqrt{2T \log N}$, actually holds for all $T$ (not just $T\leq T_0$) regardless of $N_0$ and the $\Delta_i$’s. This is stated in the proof (line 633 in the supplementary material), but we will update the theorem statement in the main body to make this explicit.
> > >
> > > ### Addressing the first point
> > >
> > > For a fixed value of $T$, as $\Delta_i$ decreases there is a transition point at $\Delta_i \propto \sqrt{(\log N) / T}$, after which $T < T_i \leq T_0$ by the definition of $T_i$ in terms of $\Delta_i$. At this transition point for $\Delta_i$, the bound transitions to the adversarial phase ($T\leq T_0$) where we bound regret by $\sqrt{T \log N}$. This transition (in $\Delta_i$) is smooth since, for a given T, it occurs exactly when $(\log N)/\Delta_i \propto \sqrt{T\log N}$ (and hence the two bounds are of the same order).
> > >
> > > In fact, our proof provides a more fine-grained perspective. In the proof of Theorem 2 (specifically, line 641 in the supplementary material), we state a more refined regret bound in terms of the values $N_0^{(t)}$ (defined in lines 214-218). Heuristically, these represent the number of experts that have not yet been identified as ineffective by round $t$ (that is, the $N_0$ effective experts plus all ineffective experts $i$ with $t \leq T_i$). As the $\Delta_i$ vary, the $T_i$ vary, so the sequence of $N_0^{(t)}$ changes accordingly, and thus the value of the regret bound changes smoothly in this quantity as well. If there were $K$ effective experts, and the smallest $\Delta_i$ goes to exactly zero, $N_0^{(t)}$ no longer converges (in $t$) to the original value of $N_0=K$, but instead converges (in fact, it is exactly equal for all rounds after $T_0$) to the new value $N_0=K+1$.
> > >
> > > We are happy to add an explicit theorem statement in Appendix B (pointed to by Remark 1) to highlight these transition points more clearly, while preserving the simplicity of only considering $\Delta_0$ in the main body as suggested by another reviewer.

---

> > > > ### Comment · Reviewer_FaPw · 2021-08-19
> > > > **Questions addressed**
> > > >
> > > > Great. Thanks for the explanation. Please do add these discussions in the revision.

---

### Official Review · Reviewer_syJt · 2021-07-13

**Rating:** 7
**Confidence:** 3

**Summary:**

In this paper, the authors tackle the problem of obtaining low semi-adversarial and quantile regret for the problem of prediction with expert advice with possibly uncountably many experts. To this end they develop a general technique to obtain methods for the problem based on the (measure-theoretical) FTRL meta-algorithm. They show that appropriate choices of regularizers can yield algorithms for the semi-adversarial (with improved dependency on the number of experts) and quantile settings (asymptotically matching state of the art). Finally, they give a lower-bound on the quantile regret, showing that their results are minimax optimal.

**Limitations And Societal Impact:**

Yes.

**Main Review:**

# Strengths

- The technique used (FTRL with uncountably many experts and a clever general technique to pick regularizers) is elegant, general, and effective;
- The lower-bound for the quantile setting, although not very well advertised, is a solid contribution to the field;

# Weakenesses

- The presentation of the paper is not optimal: it is at times hard to follow, some technical statements do not give much information to the reader, and the lack of examples (or at least their bad distribution through the paper) makes it hard to properly interpret some results.
- The measure theoretical generality is interesting, but the paper lacks a discussion of why such generality is actually worth the effort;
- The theoretical improvements on the regret seem marginal (although the experiments show that the improvements can be substantial even for a small number of experts). This by itself is not a serious weakness (incremental research can still be good and is important), but the lack of discussion on the interpretation of the result makes this weigh negatively in the paper. They seem to claim sometimes that their semi-adversarial regret bounds (Thm 2) are minimax optimal, but it is not clear if it is true;
- There is a "hidden" section: there are experiments in the appendix, but they are never mentioned in the main paper. And the big difference between the FTRL-CARL and previous algorithms is poorly discussed;

# Summary of the decision

 To fix some of the weaknesses the authors need to change a lot on the paper. Although I believe there are interesting contributions in the paper, it would deserve another round of reviews after the proposed changes, so I don't think it can be fixed for the camera-ready deadline. Of course, this decision is not necessarily final and can be changed based on the authors' response.

# Detailed discussion

I really enjoyed the techniques used, but I had a hard time going through the paper. The two main aspects that make the paper hard to read are the lack of well-placed examples and the lack of discussions of the results. For example, on section 2 it would have been great to see that we can get the Shannon entropy in eq. (1) (this is done only by the end of the paper). Also, the paper has almost no examples that exploit the measure-theoretical generality, so it is not clear why the authors go through this route. Even if the reason is purely theoretical novelty and generality, I feel this should have been better discussed.

One section that was poorly discussed was section 3, which apparently should be one of the main sections of the paper. Lemma 1 and Theorem 1 are hardly interpretable and the authors do not have any discussion trying to interpret these statements. For example, it is not clear how hard the iterate given by Lemma 1 is to compute! (The authors do claim that is a "simple task of finding the root of a 1-dimensional equation", but in the *last* section of the paper finally say it is hard to compute it. It then makes the reader wonder how useful these techniques really are. I still think these are very valuable, even if sometimes only theoretically, but the lack of up-front discussion makes the results very confusing, and leaving mention of hardness only to the end is fishy).   As another example, it is really hard to interpret the second summation in Theorem 1 and the authors do not try to explain when the summation can be reasonably bounded. In fact, nowhere in the main body of paper the authors actually use Theorem 1 (and hardly use Lemma 1), so these could have been easily deferred to the appendix, leaving more space for discussion.

For the semi-adversarial regret bounds on section 4.1, the case $T > T_0$ could have been deferred to the appendix (it could have been at least stated in a simplified way probably). It takes a lot of valuable space and there is little to no information we can obtain from it. Certainly seeing how the derivation of simpler bounds such as the one on Pg. 5 or on remark 1 follow from these general bound would help, but as it is it just takes valuable space from the authors. As an example of what could have been discussed, the regret bound for $T \leq T_0$ has no dependence on $\Delta_0$ or the additional $\log N$ term. Could this explain the excellent performance seen in the experiments? Also, it is not totally clear when $T_0$ is big or small, making it harder to understand when the first tighter bound holds. Also, although the authors do discuss some of the intuition behind $h_C$, it would have been great to know that it is a by-product of Lemma 1 from the start.

For root KL and quantile regret bounds, it is again hard to interpret the results. Corollary 2 is the main regret bounds but it is really hard to know what are some reasonable choices of $f$ so that constants $k_1$ and $k_2$ exist. A formal statement of the $\epsilon$-regret bounds the authors can get from Corollary 2 would have been extremely helpful. The lower-bound is a solid and good contribution, but it is hard for the reader to see if Corollary 2 indeed yields minimax optimal $\epsilon$-regret. Maybe this is exactly the point of Example 3, but the authors do not make it clear.

In fact, the examples from section 4.2.3 would have been extremely helpful earlier in the paper (even with the instantiations of the regret bounds deferred to the end). I strongly suggest bringing some of these examples earlier in the paper. Even in the examples some parts are not well explained (on example 2 it is not clear when/if $k_{t+1}^*$ is efficiently computable, and in fact it seems it can be relatively hard to compute).

Finally, the experiments section is hidden and I found it out almost by accident. Even being in the appendix (and thus, not bounded by space), this sections still lacks discussion of the results. It is not clear why the difference of FTRL-CARL is so big if compared to META-CARE. Like I mentioned, it might be because of the case $T \leq T_0$ or even better constants, but the authors do not discuss these aspects.

# Comments Post Rebuttal

The authors provided concrete changes (with thorough discussion) that will improve the presentation. Furthermore, they showed I had misunderstood the contribution of one of the results, and the committee discussion helped me better appreciate the results, so I'm increasing my score to 7.

I still think the papers deserves and needs a better presentation, even more so for NeurIPS' wide audience. I again encourage the authors to apply the proposed changes and use the extra space to aid in the papers readability and to help the reader understand each of the results (some results are hard to parse for an outsider like me).

And thanks a lot for the authors for the very solid and thorough rebuttals!

**Time Spent Reviewing:**

6

---

> ### Author Response · Authors · 2021-08-10
> **Response to reviewer syJt [Part 2 of 2]**
>
> [Begin Part 2 of 2]
>
> # Remaining Technical Concerns and Questions
>
> This second post contains the responses to the remaining technical questions and concerns that the reviewer raised.
>
> ## Addressing: *“It is not clear why the authors go through this route [of measure-theoretic generality].”*
>
> To handle continuous classes of experts, it is necessary to use Radon-Nikodym derivatives to characterize the “weight vector”. A natural motivating example, which we have not discussed extensively in the paper due to space constraints, is the setting of misspecified Bayesian inference. Recent work [Gv17, GM20] has demonstrated that optimal Bayesian prediction with a misspecified likelihood requires using a tempered likelihood, which has been well-studied (e.g., [Zha06]). Bayesian inference corresponds exactly to FTRL in the continuous case with the Shannon entropy, and the learning rate in FTRL exactly corresponds to the tempering parameter (i.e., the learning rate must be strictly smaller than one under misspecification). Our techniques provide a method to study these techniques even without the IID assumption on the data.
>
> An extension of this idea is to move beyond the Shannon entropy for generalized Bayes procedures, as discussed by [Alq21]. Our analysis furthers this extension, providing novel regret bounds. For instance, in Section 4.2.5, we provide regret guarantees for competing against the terminal posterior distribution in advance when we predict using the root-log regularizer; this is a novel contribution to obtain such results without knowledge of the terminal posterior to tune the learning rate. Even in the discrete setting, to obtain KL-regret guarantees with non-uniform priors it is necessary to use likelihood ratios, which we note required nearly as much effort as rigorously handling the continuous case.
>
> Finally, the generality of Theorem 1 (especially expressing it in terms of Radon-Nikodym derivatives or likelihood ratios) makes it clear how to select the regularizer to achieve sqrt(KL) regret bounds using FTRL, and without such a result it was unclear that it was even possible to construct FTRL regularizers for KL regret bounds. We suspect that this is why achieving such bounds with FTRL had eluded the researchers in the past. We highlight that reviewer FaPw noted:
> > Overall this paper makes solid contributions to a fundamental problem, and I support acceptance. What I particularly like is achieving the KL-type regret with an FTRL approach, different from all existing algorithms (NormalHedge, AdaNormalHege, Squint, Coin-betting, etc.). I believe at least several researchers have tried to do so but failed.
>
> ## Addressing: *“Lemma 1 and Theorem 1 are hardly interpretable.”* and *“Nowhere in the main body of the paper the authors actually use Theorem 1 (and hardly use Lemma 1).”*
>
> The interpretation of these results will be greatly aided by moving the examples earlier as suggested. Theorem 1 is a general result that we expect can be used to prove regret bounds for a variety of settings under the general FTRL framework we describe. We argue its placement in the main body (rather than the appendix) is justified precisely because it is the starting point for the multiple specific regret bounds we provide in the paper. Corollary 1 immediately demonstrates the utility of Theorem 1 by demonstrating how to obtain data-dependent variance bounds from the generic integral term that appears in Eq. (3). Furthermore, we highlight that the functional form of Theorem 1 makes it clear how to select the regularizers for both the quantile regret and semi-adversarial paradigms. Specifically, in the former, the terms in summation in Eq. (3) are balanced to cancel, while in the latter, the terms in summation in Eq. (3) are balanced to contribute the same order to the regret. These two cases correspond to the relationship $f\cdot f” \approx \pm 1$ of Eq. (4).
>
> ## Addressing: *“It is not clear how hard the iterate given by Lemma 1 is to compute!”*
>
> For finite expert classes, solving the normalizing equation provided by Lemma 1 to a given precision is essentially the same difficulty as normalizing the weights for the classical Hedge algorithm. For example, for FTRL-CARL the range of the normalizing constant scales with $\log N$, and solving this using the bisection method would require only $O(\log\log N)$ times more computation than normalizing Hedge. For uncountable expert classes, many of the general algorithms for prediction with expert advice are not easily modified to apply, and the standard analyses of regret bounds rely heavily on a finite $N$. In this setting, solving the FTRL normalizing equation is essentially as difficult as normalizing the posterior distribution for Bayesian inference. That is, often computationally intractable, yet regularly studied for its theoretical properties. Extending approximation techniques for Bayesian inference to approximate the solution to this optimization problem is an interesting question for future work.
>
> ## Addressing: *“For the semi-adversarial regret bounds on section 4.1, the case T>T_0 could have been deferred to the appendix.”*
>
> We believe that the reviewer has misinterpreted what part of the bound is important to characterize the minimax rate of regret accumulation, and consequently what part is important for comparison with existing literature to assess significance, since **results in this field focus almost exclusively on the $T>T_0$ setting.** For example, the well known “minimax optimal regret bound” in the stochastic-with-a-gap case, $(\log N)/\Delta$, is only minimax optimal for $T$ sufficiently large. For $T<T_0$, our simplified bound recovers the worst-case adversarial regret bound. For $T>T_0$, we see improvements from $\sqrt{T \log N}$ to $\sqrt{T \log N_0}, which as the experiments demonstrate can be quite significant.
>
> ## References
>
> [Alq21] Alquier, P. (2021). “Non-exponentially weighted aggregation: regret bounds for unbounded loss functions”. Proceedings of the 38th International Conference on Machine Learning.
>
> [BNR20] Bilodeau, B., Negrea, J., & Roy, D. M. (2020). Relaxing the IID Assumption: Adaptively Minimax Optimal Regret via Root-Entropic Regularization. arXiv preprint arXiv:2007.06552.
>
> [GM20] Grünwald, P. & Mehta, N. (2020). “Fast Rates for General Unbounded Loss Functions: From ERM to Generalized Bayes”. Journal of Machine Learning Research, 21, 1-80.
>
> [Gv17] Grünwald, P. & van Ommen, T. (2017). “Inconsistency of Bayesian Inference for Misspecified Linear Models, and a Proposal for Repairing It”. Bayesian Analysis, 12, 1069-1103.
>
> [MG19] Mourtada, J. & Gaïffas, S. (2019). On the optimality of the Hedge algorithm in the stochastic regime. Journal of Machine Learning Research, 20(83), 1-28.
>
> [Zha06] Zhang, T. (2006). “Information-Theoretic Upper and Lower Bounds for Statistical Estimation”. IEEE Transactions on Information Theory, 52, 1307-1321.
>
> [End Part 2 of 2]

---

> > ### Comment · Reviewer_syJt · 2021-08-20
> > **Thanks for the thoroughly addressing my concerns**
> >
> > Thanks a lot for the authors to addressing so thoroughly my concerns. Although my review was long (if I had more time it could have been shorter), the authors were successful in addressing my questions quite objectively. I will probably raise my score, I just want to complete the discussion with the other reviewers first.
> >
> > But I have a couple of comments/suggestions for the authors to consider based on your reply and on the other reviews:
> >
> > 1. I suggest you **do not bring the experiments to the main body**. It was odd to find the experiments in the appendix without any prior mention of them, but I do not think they add much to the paper. You should mention that there are experiments, but the extra-space for the camera ready version should be focused on improving the discussion of your results. You did an amazing job of doing so in the rebuttals of the reviews, and as much of this discussion as possible should go to the camera-ready version, if the paper is accepted.
> >
> > 2. Keep in mind NeurIPS is a broad conference. So discussing why the case $T > T_0$ is important on Section 4.1 is vital. In fact, I only suggested so because I had a hard time interpreting the meaning of the terms in the summations. Reading the other reviews and the rebuttals helped be both interpret the result better and appreciate its significance. So again, use the extra-space to aid in the interpretation and to claim significance of your results.
> >
> > 3. You mentioned on your rebuttal the comment in FaPw's review. It sounds interesting. If you can show to the average reader that this is the case, this would aid a lot the reader in placing the contribution of the paper among previous work.

---

> > > ### Author Response · Authors · 2021-08-23
> > > **Responses to Additional Comments/Suggestions from syJt**
> > >
> > > Thank you for your follow up questions.
> > >
> > > 1 & 2: We like these suggestions and will make sure the experiments are mentioned in the body, and that the extra discussion we’ve added through the rebuttals appears in the paper to provide more context to the reader.
> > >
> > > 3: As far as we know, that others have tried to obtain sqrt-KL regret bounds using FTRL is mostly folklore. We personally know of past attempts by two researchers based on personal correspondence. We do not know the specific basis of the other reviewer's statement that sqrt-KL bounds via FTRL have been attempted before without success. Unfortunately, people don’t often report attempts they've made that failed.
> > >
> > > While racking our brains to think of hard evidence for the claim that others have tried, we recalled two COLT open problems that could be interpreted as evidence that others have considered this problem important and that people have tried to solve it. These open problems don’t explicitly ask for sqrt-KL regret bounds via FTRL, but their formulations can be interpreted as being based on an underlying common conjecture that such an approach is possible. This interpretation is based on the fact that 1) both open problems are essentially tied to getting regret bounds in terms of local norms at the same time as getting sqrt-KL regret bounds, and 2)  local norms typically arise from FTRL analyses.
> > >
> > > The first of these open problems, stated in [Fre16], pertains to prediction with expert advice for losses on an unknown scale, and asks for a regret bound that depends on the predictive variances. The second, stated in [FKL20], pertains to model selection for contextual bandits and asks for sqrt-KL regret bounds for model selection with bandit feedback. In both of these cases, the FTRL analysis of Hedge yields local-norms equal to predictive variances but KL terms, not sqrt-KL terms. These open problems ask whether one can improve the KL term to sqrt-KL, while maintaining the local norm contributions. We can add a paragraph describing the relationship between the present work and those open problems in the final version (based on the common goal of combining sqrt-KL regret bounds with local norms), if the reviewer feels this would help the reader place the contribution of this work.
> > >
> > > The present work does not resolve these open problems. The gaps remaining to resolve the open problem in [Fre16] are that 1) the predictive "variances" resulting from our analysis would be based on a *reweighted* predictive distribution (reweighted according to $1/f{''}$), and 2) instability introduced by adaptive learning rates would need to be addressed. We do not know of a solution to these two problems at this point in time. The gap remaining to resolve the open problem in [FKL20] is, again, tied to the predictive variances resulting from our analysis; in the bandit case, the expected predictive variance of the importance weighted losses for Hedge can be bounded by the number of arms, however the reweighting by $1/f{''}$ affects this, and presents new challenges for proving their desired result. Thus, our work can be seen as a step towards the solution of these problems, but they both remain open.
> > >
> > >
> > > [Fre16] Freund, Y. (2016). Open Problem: Second order regret bounds based on scaling time. In Conference on Learning Theory (pp. 1651-1654). PMLR.
> > >
> > > [FKL20] Foster, D. J., Krishnamurthy, A., & Luo, H. (2020). Open problem: Model selection for contextual bandits. In Conference on Learning Theory (pp. 3842-3846). PMLR.

---

> > > > ### Comment · Reviewer_syJt · 2021-08-29
> > > > **Review updated**
> > > >
> > > > Thanks for the authors for the very detailed and thorough reviews! I've updated my review accordingly, and enforcing my suggestion to improve the presentation (which was the weakest point of the paper, according to the committee discussion).

---

> ### Author Response · Authors · 2021-08-10
> **Response to reviewer syJt [Part 1 of 2]**
>
> [Begin Part 1 of 2]
>
> We thank the reviewer for their time and detailed comments.
>
> ## Summary
>
> We believe the reviewer’s main concerns can be summarized as follows:
> 1. The reviewer questions the significance of our regret bounds.
> 2. The reviewer finds our general theorems (Theorem 1 and Lemma 1) confusing to interpret.
> 3. The reviewer finds our specific regret bounds (Theorem 2 and Corollary 2) difficult to contextualize with the examples and consequently difficult to identify the significance.
>
> The reviewer claims that in light of these concerns, the changes required to fix them are too numerous and would require further review. In what follows, we provide a detailed response to their comments with substantial elaboration to clarify for the reviewer. Moreover, we **specifically refute the claim that the necessary changes are too numerous**. In particular, much of our elaboration below is not content to be added to the paper, and is included in our response to directly address the reviewer’s questions and the reviewer misinterpreting what aspects of our regret bound are important in determining that FTRL-CARL achieves the minimax rate of regret. Consequently, most of the actual changes needed in the paper are organizational (as suggested by the reviewer, and enumerated below). We believe that the detail of the review has sufficiently warranted the detail of our response, which spans two posts.
>
> Our response is organized into three parts. First, we address the questions regarding the significance of our results. Second, we describe the minor structural changes that are required to resolve the majority of the criticisms raised by the reviewer. Third, we address the additional technical questions and concerns of the reviewer. While the changes that will be required in the paper are minor, the response includes detail commensurate to that of this thorough review.
>
> A summary of the changes that are actually required in the paper is:
> 1. Move the examples of Section 4.2.3 to before Section 3, in order to concretely state how the applications of our results can be compared with existing literature.
> 2. Move the additional settings of Section 4.2.5 to directly after the statement of Theorem 1 and Lemma 1 to motivate the generality of considering a continuum of experts (and consequently requiring the measure-theoretic notation).
> 3. Add a paragraph bridging Lemma 1 and Theorem 1 with this moved Section 4.2.5, which explains their interpretation (see the specific response below for the details of this paragraph).
> 4. State Theorem 2 in the simplified language of Corollary 2, and move the full generality of Theorem 2 to the appendix.
> 5. Move the experiments section from the appendix to the main body (last section, using the extra page provided for camera-ready), and add a brief paragraph explaining the improved performance seen in the plots.
>
> In short, this corresponds to: move 3 sections as-is, add 2 paragraphs bridging the moved sections, and move a theorem to the appendix.
>
> Our responses to the technical questions and concerns are mostly deferred to the second post. A summary of them is as follows:
> 1. Address the necessity and relevance of our measure-theoretic generality, which has also been raised (and addressed) for other reviewers.
> 2. Explain how to interpret Theorem 1 and Lemma 1.
> 3. Address the computational complexity of the formula in Lemma 1.
> 4. Address the reviewer’s questions about Theorem 2 by noting that the reviewer has misunderstood the semi-adversarial regret bounds, and in particular their dependence on the term T0, which is standard in classical results from the stochastic setting of prediction with expert advice.
>
> # Defending Significance of our Results
>
> ## Addressing: *“The theoretical improvements on the regret seem marginal.”*
>
> The reviewer does not provide direct justification for this claim, which makes it difficult to directly refute. Instead, we briefly summarize our contributions, all of which are discussed in the paper. As mentioned above, these impacts are potentially obfuscated by the current organization of the paper, and moving the examples earlier will help clarify these impacts in the paper.
>
> For the semi-adversarial paradigm, existing algorithms are either provably suboptimal (if Hedge-based, as proved by [BNR20]) or it is unclear how to prove optimality (particularly for potential-based algorithms). Meta-Care of [BNR20] is adaptively minimax optimal, which is characterized using the asymptotic (in $T$) rate of regret, but we improve the finite time regret guarantees using FTRL-CARL by reducing a $(\log N)^{3/2}/\Delta_0$ term to $(\log N)/\Delta_0$. This can have a dramatic impact for smaller T.
>
> For the quantile regret paradigm, as noted by reviewer FaPw, we are the first to prove quantile regret bounds using an FTRL algorithm, which has been an open problem in the community. As this reviewer (and others) note, our lower bound demonstrating the quantile regret performance of abNormal is minimax optimal is also novel and is applicable to quantile regret studied in multiple other papers.
>
> ## Addressing: *“They seem to claim sometimes that their semi-adversarial regret bounds (Thm 2) are minimax optimal, but it is not clear if it is true.”*
>
> The matching lower bounds that characterize the minimax rate of regret for each setting in the semi-adversarial paradigm are proved and stated in [BNR20] (the lower bound for the $N_0=1$ case in that work is inherited from [MG19]); we will restate them in the paper to allow the reader to more clearly see that we match them. In particular, FTRL-CARL is *adaptively* minimax optimal, which means that it obtains the minimax optimal rate in every setting without knowing in advance what the setting is. This is a much stronger notion of optimality than is considered in classical results of this form, and is implicitly the notion of optimality that [MG19] consider (although they only have two settings between which to adapt, $N_0=1$ and adversarial, and their result for $N_0=1$ but not IID has a spurious $\log(1/\Delta_0)/\Delta_0$ term).
>
> # Changes to be Made in the Paper
>
> ## Addressing: *“To fix some of the weaknesses the authors need to change a lot on the paper ... it would deserve another round of reviews after the proposed changes, so I don't think it can be fixed for the camera-ready deadline.”*
>
> The reviewer strongly suggests a) moving the examples earlier and elaborating on them to motivate the generality of our core results and b) moving the experiments into the main body and providing more discussion on them. Both of these changes are rather minor to make due to the extra page available for the camera-ready version and the space we will save by presenting only a simplified version of Theorem 2 (as suggested in this review). **Moreover, making these two changes resolves many of the comments raised in the review, directly refuting the statement that there are too many changes to make.** We now enumerate the comments of the review that are resolved by these two changes to the ordering of sections.
>
> 1. *“The presentation of the paper is not optimal: it is at times hard to follow, some technical statements do not give much information to the reader, and the lack of examples (or at least their bad distribution through the paper) makes it hard to properly interpret some results.”*
> 2. *“It would have been great to see that we can get the Shannon entropy in eq. (1) (this is done only by the end of the paper).”*
>
>
> Example 1 corresponds to the Shannon entropy (Hedge).
>
> 3. *“It is hard for the reader to see if Corollary 2 indeed yields minimax optimal KL-regret. Maybe this is exactly the point of Example 3, but the authors do not make it clear.”* and *“A formal statement of the regret bounds the authors can get from Corollary 2 would have been extremely helpful.”*
>
>
> Example 3 is precisely the formal statement of the regret bounds we obtain from Corollary 2 for a specific instantiation of our abNormal algorithm.
>
> 4. *“In fact, the examples from section 4.2.3 would have been extremely helpful earlier in the paper (even with the instantiations of the regret bounds deferred to the end).”*
>
> ## Addressing: *“Certainly seeing how the derivation of simpler bounds [for Section 4.1] such as the one on Pg. 5 or on remark 1 follow from these general bounds would help, but as it is it just takes valuable space from the authors.”*
>
> The statement of Theorem 2 is complicated by considering a separate effective gap for each expert; as suggested, we will simplify the statement in the main body to only a single effective gap, allowing for more direct comparison to [BNR20] while deferring the more detailed bound to the appendix. The more detailed bound shows a smooth transition from the $\sqrt{T \log N}$ phase to the $\sqrt{T \log N_0}$ phase via the $N^{(t)}$ terms we define.
>
> ## Addressing: *“The experiments section is hidden… [and] lacks discussion of the results.”* and *“The regret bound for T<= T_0 has no dependence on \Delta_0 or the additional logN term. Could this explain the excellent performance seen in the experiments?”*
>
> We regret that the experiments section was not mentioned in the main body; this was an oversight due to space constraints. Using the extra page available for the camera-ready, we will move the experiments into the main body and provide more discussion. For $N_0>1$, FTRL-CARL improves significantly on Mete-CARE by reducing a $(\log N)^{3/2}/\Delta_0$ term to $(\log N)/\Delta_0$. This is especially important when $T$ is not exponentially larger than $N$. In the plots, $T_0$ corresponds to roughly 700, so in fact the majority of the plots (and the demonstrated improvement) occurs for $T>T_0$. The simplification of the statement of Theorem 2 described above will also make this dependence clearer in the experiments.
>
> [End Part 1 of 2]

---

### Official Review · Reviewer_6vq4 · 2021-07-16

**Rating:** 6
**Confidence:** 2

**Summary:**

This paper considers the problem of prediction with expert advice,
and focuses on quantile regret bounds that depend on the comparator distribution
and semi-adversarial paradigm,
which is an intermediate setting between stochastic one and adversarial one.
In the paper,
minimax optimal regret bounds for both paradigms are shown.
This bound is achieved by follow-the-regularized-leader (FTRL) algorithm with a newly designed root-logarithmic regularizers.

**Limitations And Societal Impact:**

I think limitations and societal impact have been well discussed.

**Main Review:**

Originality:

The originality of the contributions in the semi-adversarial paradigm seems somewhat limited.
The method of analysis appears to be almost the same as that in this literature [BNR20],
in which the semi-adversarial paradigm has been introduced.
In addition, the semi-adversarial setting with $N_0 = 1$ has already been treated in [GSv14] (as Assumption 1 in [GSv14] holds), and it has already been proved that an upper bound of $O(\log N / \Delta_0)$ can be achieved in this case.
We would like to note that the semi-adversarial setting is also a special case of adversarial regimes with a self-bounding constraint in [ZS21].

Quality:

As far as I can see, no technical errors have been found.
I think the authors are careful and honest about evaluating both the strengths and weaknesses of their work.

Clarity:

The technical part is clearly described.
However,
the comparison with existing studies is not very clear.
In particular,
I did not know if the proposed algorithm was more effective than the standard Hedge algorithm (in [MG19]) and algorithms with second-order regret bounds (in [GSv14], [CMS07])
in semi-adversarial paradigm.

Significance:

The theoretical results on quantile regrets will have a certain impact.
In particular, as far as I know, this is the first time that a lower bound for for quantile regret on finite expert classes has been given.
Compared to this, the impact of the results on semi-adversarial setting seem to be somewhat limited, for the reasons mentioned above.



[ZS21]: Zimmert, J., & Seldin, Y. (2021). Tsallis-INF: An Optimal Algorithm for Stochastic and Adversarial Bandits. J. Mach. Learn. Res., 22, 28-1.

**Time Spent Reviewing:**

5

---

> ### Author Response · Authors · 2021-08-10
> **Response to reviewer 6vq4  [references]**
>
> ## References
>
> [ACF02] Auer, P., Cesa-Bianchi, N., & Fischer, P. (2002). “Finite-Time Analysis of the Multiarmed Bandit Problem”. Machine Learning, 47, 235-256.
>
> [BNR20] Bilodeau, B., Negrea, J., & Roy, D. M. (2020). Relaxing the IID Assumption: Adaptively Minimax Optimal Regret via Root-Entropic Regularization. arXiv preprint arXiv:2007.06552.
>
> [CMS07] Cesa-Bianchi, N., Mansour, Y., & Stoltz, G. (2007). “Improved Second-Order Bounds for Prediction with Expert Advice.” Machine Learning, 66, 321-352.
>
> [GSv14] P. Gaillard, G. Stoltz, & T. van Erven (2014). “A second-order bound with excess losses”. Conference on Learning Theory, 176-196.
>
> [MG19] Mourtada, J. & Gaïffas, S. (2019). On the optimality of the Hedge algorithm in the stochastic regime. Journal of Machine Learning Research, 20(83), 1-28.
>
> [ZS21] Zimmert, J., & Seldin, Y. (2021). Tsallis-INF: An Optimal Algorithm for Stochastic and Adversarial Bandits. Journal of Machine Learning Research, 22(28), 1-49.

---

> ### Author Response · Authors · 2021-08-10
> **Response to reveiwer 6vq4**
>
> We thank the reviewer for their time and comments.
>
> First, we appreciate that the reviewer has noted the following:
>
> >As far as I know, this is the first time that a lower bound for quantile regret on finite expert classes has been given.
>
> We are also unaware of any existing lower bounds for quantile regret, and believe our result is a strong contribution to contextualize the existing upper bounds in this setting (including our own).
>
> ## Summary
>
> The reviewer’s concerns can be summarized as questioning the originality and significance of the results in the semi-adversarial paradigm. We address these in detail in what follows by a) demonstrating the papers that the reviewer claimed subsume the semi-adversarial paradigm are either special cases of the semi-adversarial paradigm or constitute a different type of constraint (cumulative vs. instantaneous); b) reiterating the contribution of our regret bound for FTRL-CARL over the regret bound for Meta-Care from [BNR20]; and c) providing an argument for suboptimality of certain existing algorithms in the semi-adversarial paradigm, which is not in the paper due to space constraints but provides more context for the reviewer.
>
> ## Addressing: *“The originality of the contributions in the semi-adversarial paradigm seems somewhat limited.”*
>
> The questions of originality and significance for the semi-adversarial results focus only on the setting where there is a single “good” (effective) expert, which a) misses the majority of the settings in the semi-adversarial paradigm (i.e., all of $N_0>1$) and b) misses that our algorithm FTRL-CARL is *adaptively* minimax optimal. In particular, the goal is not simply to obtain the minimax optimal rate in any one setting (indeed, classical results do this for the IID setting and the fully adversarial setting separately), but to obtain the minimax optimal rate in every setting without knowing in advance what the setting is. This is a much more stringent goal, and is not attempted by [GSv14] or [CMS07]. It is achieved at the two endpoints of $N_0=1$ and $N_0=N$ by [ZS21] (in the bandit setting), and as we (and [BNR20]) cite this was already done in the full-information setting by [MG19], yet their algorithm (Hedge) is provably not adaptively minimax optimal for the full spectrum of semi-adversarial settings (see [BNR20]).
>
> The definition of adaptively minimax optimal is in [BNR20], so we sympathize with the potential confusion this caused for the reviewer. However, it is implicitly used in both [ZS21] and [MG19] (and, in fact, all work on stochastic and adversarially optimal methods), and is standard in statistics literature on smoothness adaptation. In light of this, we strongly disagree with the statement that our results are treated by existing literature. However, for completeness and to avoid such confusion for readers, we will highlight the notion of adaptivity we consider in the final paper.
>
> ## Addressing: *“The semi-adversarial setting is also a special case of adversarial regimes with a self-bounding constraint in [ZS21].”* and *“In addition, the semi-adversarial setting with $N_0=1$ has already been treated in [GSv14].”*
>
> In fact, **the semi-adversarial paradigm is not a special case of either [GSv14] or [ZS21].**
>
> Assumption 1 of [GSv14], which requires that the losses are independent and have a gap in the means, actually corresponds to a special case of the semi-adversarial paradigm when N0=1. The semi-adversarial paradigm does not assume independence, and the conditional distribution played by the adversary/environment at each time can depend on the history of the game, even if there is a single good expert in expectation. The $N_0=1$ case is handled in [MG19] (although they have a spurious $\log(1/\Delta)/\Delta$ term in their regret bound), and can be viewed analogously (with the caveats on pseudo-regret below) to the self-bounding constraint regime of [ZS21]. However, the $N_0=1$ case of [MG19] and the self-bounding constraint of [ZS21] are not exactly the same, since the former is an instantaneous constraint (the loss distribution on each round is constrained) while the latter is a cumulative constraint at the terminal time only. Moreover, [MG19] only handles $N_0 \in \\{1,N\\}$. Even if the settings are the same in the $N_0=1$ case, the algorithm therein is provably not adaptively minimax optimal for $N_0 \in \\{1,2,3,...,, N-1, N\\}$ (as shown in [BNR20]). Thus, even though the settings are comparable when $N_0 \in \\{1,N\\}$, the more general adaptivity results are not, with our result (adaptive minimax optimality) being much stronger. As we argue below, the same holds for [GSv14] as well.
>
> For $N_0>1$, the semi-adversarial paradigm is not subsumed by the self-bounding constraint regime of [ZS21]. First, as the authors of that work state, it is unclear how to extend the analysis in [ZS21] to a self-bounding constraint with more than one best expert. Because of this, both the self-bounding constraint and the corresponding analysis in [ZS21] cannot be used in the case when $1<N_0<N$. Second, the self-bounding constraint is a sufficient condition on *cumulative* losses for proving *pseudo-regret* bounds, while the semi-adversarial paradigm describes constraints on *instantaneous* losses.
>
> Finally, [ZS21] studies pseudo-regret while we study expected regret. These can be extremely different: there are settings where expected regret is lower bounded by $\sqrt{T \log N}$ yet the pseudo-regret is $0$, and thus the techniques of [ZS21] may not be not sufficient to bound expected regret (even in the case of $N_0=1$).
>
> **For the benefit of the AC and the reviewer, we now summarize the impacts of our semi-adversarial regret bound, which are also enumerated in the paper:**
> 1. We improved the regret upper bound in the $N_0>1$ case by reducing the power of $(\log N)$ from $3/2$ to $1$ for the term where it is multiplied by $1/\Delta_0$. The dramatic impact of this improvement is demonstrated even for small $N$ in our experiments, which we will move into the main body on the suggestion of another reviewer.
> 2. We characterized the dependence of regret on individual gaps, as opposed to only the smallest gap. This extends the semi-adversarial analysis of [BNR20] (and the stochastic-with-a-gap setting of [GSv14]) to more closely match known results from the bandit literature (e.g. [ACF02]). Even for classical stochastic bandits, lower bounds that depend on the individual gaps are not known, and thus fully characterizing such a refined notion of adaptive minimax optimality remains open; due to space constraints, we did not discuss this last detail in the paper, but will add a 1-2 sentence description.
> 3. We have smoothly characterized the full trajectory of the regret from the phase where no experts are identified as ineffective to the phase where the learner has identified the exact effective expert set, and each ineffective expert is identified individually in order of how identifiable they are from the effective experts.
>
> ## Addressing: *“I did not know if the proposed algorithm was more effective than the standard Hedge algorithm (in [MG19]) and algorithms with second-order regret bounds (in [GSv14], [CMS07]) in semi-adversarial paradigm.”*
>
> First, note that in [BNR20] it was proved that Hedge is suboptimal for $N_0$ not in $\{1,N\}$, which we point out at the beginning of Section 4. Next, we emphasize that we make no claims that FTRL-CARL is the only optimal algorithm for the semi-adversarial setting, although we do emphasize the benefits of being able to achieve multiple goals (semi-adversarial and quantile regret) using FTRL. The two specific papers mentioned both give algorithms similar to Hedge with an adaptive learning rate.
> 1. [CMS07] is essentially Hedge with an adaptive learning rate shared by all experts. The example in [BNR20] that shows Hedge is suboptimal when $N_0$ is not in $\\{1,N\\}$ would also show that Hedge with this form of adaptive learning rate shared by all experts would be sub-optimal as well, because the cumulative squared losses (the data-dependent quantity that is used in their learning rate) will be (essentially) a constant multiple of $T$ in that example.
> 2. [GSv14] is more subtle, since they allow a different learning rate for each expert. It is possible that the learning rates could be chosen to essentially yield FTRL-CARL using a similar choice to the heuristic derivation we provide, although it is not clear whether the same analysis could be used. Further, the recommended learning rate (Corollary 4 of their paper) would not achieve this since it uses $\sqrt{\log N}$ for all experts, as opposed to $\sqrt{H(w)}$ as in FTRL-CARE ([BNR20]) or $\sqrt{\log(1/w_i)}$ for expert $i$ as in FTRL-CARL (present work). Using the suggested learning rates they provide in [GSv14], the example from the lower bound in [BNR20] would make that algorithm behave essentially like Hedge again; that is, if we pick $N_0$ big enough and all the losses are $0$ or $1$, the average loss with respect to the Adapt-ML-Prod weights will be roughly $1/2$ for all t large enough. Thus, the lower bound from [BNR20] would essentially yield a lower bound for Adapt-ML-Prod with the learning rate in Corollary 4 of [GSv14] as well.
>
> There are dozens of other candidate algorithms that one might ask if they are adaptively minimax optimal for the semi-adversarial paradigm, however for many of these (especially potential based algorithms) it is not clear how one would prove a positive result. The natural way to prove a negative result would be as above, to reduce it to the suboptimality of Hedge if possible, since many of the candidates can also be viewed as modifications to Hedge. Based on this, we will add a paragraph describing that Hedge and adaptive-Hedge-like algorithms in the literature do not typically achieve adaptively minimax optimal semi-adversarial regret.

---

### Official Review · Reviewer_o5hQ · 2021-07-17

**Rating:** 7
**Confidence:** 2

**Summary:**

The paper studies Follow the Regularised Leader algorithms for the Hedge settings.

The paper starts with an overview of FTRL in adversarial settings. A frighteningly general setup is described and Theorem 1 for the adversarial regret is obtained. It is claimed that the theorem allows one to obtain bounds similar to some known bounds from the literature.

Then the semi-adversarial settings are discussed (the definition of semi-adversarial covers the adversarial settings on one end of the spectrum and purely stochastic on another).

The paper presents FTRL-CARL and obtains a performance bound on it in terms of effective stochastic gaps.

Then the paper discusses bounds in terms of f-divergence and obtains a lower bound matching upper bounds.

**Ethical Concerns:**

No.

**Limitations And Societal Impact:**

Yes.

**Main Review:**

The paper is not very kind to the reader and the presentation is often very formal. For example, I understand the most important results were obtained for the case of finitely many experts. Was it really necessary to introduce the general framework with Radon-Nikodym derivatives?

It is hard for me to appreciate the importance of CARL bounds as the concept of effective stochastic gaps is not clear to me. Why are they important and how do they characterise the distribution class?

The matching lower bound is something I can understand and appreciate and find very important.

Finally, I can't help thinking that discussing semi-stochastic settings and f-divergence bounds in one paper is somewhat artificial.

**Time Spent Reviewing:**

5

---

> ### Author Response · Authors · 2021-08-10
> **Response to reviewer o5hQ**
>
> We thank the reviewer for their time and comments. We address all three comments point by point.
>
> ## Addressing: *“Was it really necessary to introduce the general framework with Radon-Nikodym derivatives?”*
>
> Yes. To handle continuous classes of experts, it is necessary to use Radon-Nikodym derivatives to characterize the “weight vector”. A natural motivating example, which we have not discussed extensively in the paper due to space constraints, is the setting of misspecified Bayesian inference. Recent work [Gv17, GM20] has demonstrated that optimal Bayesian prediction with a misspecified likelihood requires using a tempered likelihood, which has been well-studied (e.g., [Zha06]). Bayesian inference corresponds exactly to FTRL in the continuous case with the Shannon entropy, and the learning rate in FTRL exactly corresponds to the tempering parameter (i.e., the learning rate must be strictly smaller than one under misspecification). Our techniques provide a method to study these techniques even without the IID assumption on the data.
>
> An extension of this idea is to move beyond the Shannon entropy for generalized Bayes procedures, as discussed by [Alq21]. Our analysis furthers this extension, providing novel regret bounds. For instance, in Section 4.2.5, we provide regret guarantees for competing against the terminal posterior distribution in advance when we predict using the root-log regularizer; this is a novel contribution to obtain such results without knowledge of the terminal posterior to tune the learning rate. Even in the discrete setting, to obtain KL-regret guarantees with non-uniform priors it is necessary to use likelihood ratios, which we note required nearly as much effort as rigorously handling the continuous case.
>
> Finally, the generality of Theorem 1 (especially expressing it in terms of Radon-Nikodym derivatives or likelihood ratios) makes it clear how to select the regularizer to achieve sqrt(KL) regret bounds using FTRL, and without such a result it was unclear that it was even possible to construct FTRL regularizers for KL regret bounds. We suspect that this is why achieving such bounds with FTRL had eluded the researchers in the past. We highlight that reviewer FaPw noted:
> > Overall this paper makes solid contributions to a fundamental problem, and I support acceptance. What I particularly like is achieving the KL-type regret with an FTRL approach, different from all existing algorithms (NormalHedge, AdaNormalHege, Squint, Coin-betting, etc.). I believe at least several researchers have tried to do so but failed.
>
> ## Addressing: *“Why are [effective stochastic gaps] important and how do they characterise the distribution class?”*
>
> Classical results on prediction with expert advice (and the bandit variant) in the stochastic setting obtain regret bounds in terms of a Delta that quantifies how identifiable the best expert is from the rest  [ACF02]. The role of the effective stochastic gaps is to generalize this definition to the semi-adversarial setting, quantifying how identifiable the “good” (effective) experts are from the “bad” (ineffective) experts.
>
> For a more concrete interpretation, the stochastic gap is also an important quantity in hypothesis testing for the equality of two means (e.g., in the classical work [RS65]). There, the sample size needed to guarantee a minimum probability for concluding the means are different will depend on the gap between the means; more data is needed to confidently determine that two features have different means when those means are closer together. This same effect is witnessed here, where any prediction algorithm will need a sufficiently large number of examples before it is confident one expert is better than another.
>
> ## Addressing: *“Discussing semi-stochastic settings and f-divergence bounds in one paper is somewhat artificial.”*
>
> As we discuss in the paper, both of these paradigms are related in two distinct ways. First, they both correspond to “easier” notions of prediction with expert advice than the classical worst case analysis. Second, the optimal algorithms we provide for each paradigm are technically related by corresponding to FTRL with linearly decomposable regularizers satisfying the relationship described by Eq. (4).  We strongly believe this connection will enable future efforts to prove regret bounds for other “easier” paradigms using the robust framework and analysis we have developed for FTRL.
>
> ## References
>
> [Alq21] Alquier, P. (2021). “Non-exponentially weighted aggregation: regret bounds for unbounded loss functions”. Proceedings of the 38th International Conference on Machine Learning.
>
> [ACF02] Auer, P., Cesa-Bianchi, N., & Fischer, P. (2002). “Finite-Time Analysis of the Multiarmed Bandit Problem”. Machine Learning, 47, 235-256.
>
> [GM20] Grünwald, P. & Mehta, N. (2020). “Fast Rates for General Unbounded Loss Functions: From ERM to Generalized Bayes”. Journal of Machine Learning Research, 21, 1-80.
>
> [Gv17] Grünwald, P. & van Ommen, T. (2017). “Inconsistency of Bayesian Inference for Misspecified Linear Models, and a Proposal for Repairing It”. Bayesian Analysis, 12, 1069-1103.
>
> [RS65] Robbins, H. & Starr, N. (1965). “Remarks on Sequential Hypothesis Testing”. University of Minnesota Technical Report No. 68.
>
> [Zha06] Zhang, T. (2006). “Information-Theoretic Upper and Lower Bounds for Statistical Estimation”. IEEE Transactions on Information Theory, 52, 1307-1321.

---

### Decision · Program_Chairs · 2021-09-27

**Decision:**

Accept (Poster)

**Comment:**

The reviewers unanimously support acceptance of the paper. The writing of the paper requires improvement and I hope that the authors will implement changes promised to the reviewers.